evolution, palaeontology

Cambrian, Euarthropoda, Opabiniidae, phylogenetics, treespace, Wheeler Formation

**Authors for correspondence:**
Stephen Pates
e-mail: sp587@cam.ac.uk
Joanna M. Wolfe
e-mail: jowolfe@g.harvard.edu

†These authors contributed equally to this study.

### PUBLISHING

# New opabiniid diversifies the weirdest wonders of the euarthropod stem group

Stephen Pates[1,2,†], Joanna M. Wolfe[1,†], Rudy Lerosey-Aubril[1], Allison C. Daley[3] and Javier Ortega-Hernández[1]

[1]Museum of Comparative Zoology and Department of Organismic and Evolutionary Biology, Harvard University, 26 Oxford Street, Cambridge, MA 02138, USA
[2]Department of Zoology, University of Cambridge, Downing Street, Cambridge CB2 3EJ, UK
[3]Institute of Earth Sciences, University of Lausanne, Géopolis, 1015 Lausanne, Switzerland

SP, 0000-0001-8063-9469; JMW, 0000-0001-6708-8332; RL-A, 0000-0003-2256-1872;
ACD, 0000-0001-5369-5879; JO-H, 0000-0002-6801-7373

Once considered 'weird wonders' of the Cambrian, the emblematic Burgess Shale animals *Anomalocaris* and *Opabinia* are now recognized as lower stem-group euarthropods and have provided crucial data for constraining the polarity of key morphological characters in the group. *Anomalocaris* and its relatives (radiodonts) had worldwide distribution and survived until at least the Devonian. However, despite intense study, *Opabinia* remains the only formally described opabiniid to date. Here we reinterpret a fossil from the Wheeler Formation of Utah as a new opabiniid, *Utaurora comosa* nov. gen. et sp. By visualizing the sample of phylogenetic topologies in tree-space, our results fortify support for the position of *U. comosa* beyond the nodal support traditionally applied. Our phylogenetic evidence expands opabiniids to multiple Cambrian stages. Our results underscore the power of treespace visualization for resolving imperfectly preserved fossils and expanding the known diversity and spatio-temporal ranges within the euarthropod lower stem group.

## 1. Introduction

Euarthropods (e.g. chelicerates, myriapods and pancrustaceans including insects) have conquered Earth's biosphere, comprising over 80% of living animal species [1]. Indeed, Euarthropoda has been the most diverse animal phylum for over half a billion years, documented by prolific trace and body fossil records that extend back to the early Cambrian (approx. 537 and approx. 521 Ma, respectively) [2]. As most of these early euarthropods did not possess mineralized hard parts, we rely on remarkable fossil deposits such as the Burgess Shale, which preserve soft-bodied components of ancient biotas, to reveal critical data on the extraordinary diversity, disparity and evolution of Cambrian euarthropods [3].

Two of the most peculiar Burgess Shale animals, *Anomalocaris* and *Opabinia*, illustrate the complicated history of research of many Cambrian soft-bodied taxa—a result of their unfamiliar morphologies compared to the occupants of modern oceans [4–6]. Both *Anomalocaris* and *Opabinia* possess compound eyes, lateral swimming flaps, filamentous setal structures and a tail fan [7–10]. *Anomalocaris* and its relatives, the radiodonts, are united by the presence of paired sclerotized protocerebral frontal appendages and mouthparts composed of plates of multiple sizes, forming a diverse group containing over 20 species [11–18]. Radiodonts range in age from the early Cambrian to at least the Devonian, and have been recovered from numerous palaeocontinents [12,14,19–22]. Meanwhile, the most celebrated animal from the Burgess Shale [5,23], *Opabinia regalis*, with its head bearing five stalked eyes and a proboscis, remains the only opabiniid species confidently identified and is only known from a single

quarry in the Burgess Shale. The enigmatic *Myoscolex ateles* from the Emu Bay Shale, originally described as a possible polychaete [24], was also recently proposed as a possible close relative of *O. regalis* [25]. However, the presence of morphological features supporting this latter interpretation are controversial [26], leaving its affinities uncertain.

Radiodonts and *Opabinia* are now confidently placed within the lower stem of Euarthropoda [11,23,27], following the assignment of nearly all Cambrian soft-bodied animals to stem and crown groups of modern phyla (e.g. [28]). Fossils illustrating the sequence of character evolution along the euarthropod stem lineage provide the framework for understanding the evolutionary origins of the segmented, modular exoskeleton and the specialized appendages that underpin the ecological success of this phylum [27]. The lower stem group charts euarthropod evolution from lobopodian-like ancestors with paired gut diverticulae and lobopodous limbs [29], through taxa like *Opabinia* with swimming flaps associated with filamentous gill structures [9,30,31], to radiodonts, the first to possess arthropodized appendages [11]. Deuteropoda, defined by the presence of a multisegmented head with hypostome–labrum complex and differentiated deutocerebral appendages, comprises upper stem and crown Euarthropoda [27].

Difficulties remain in interpreting the anatomical details, morphology and phylogenetic placement of exceptional Cambrian fossils. In *Opabinia*, the presence of lobopodous limbs in addition to the swimming flaps cannot be confirmed, and the architecture of the flaps and associated setal blades remains elusive [9,23,32]. The identification of plesiomorphic and apomorphic characters within the euarthropod stem lineage has required new imaging and reinterpretations of existing specimens, the discovery of new fossil material and localities, and, crucially, the improvement of phylogenetic analysis methods to evaluate alternative relationships of enigmatic taxa.

Here we redescribe a fossil specimen from the Drumian Wheeler Formation of Utah, previously described as an anomalocaridid radiodont [33]. *Utaurora comosa* nov. gen. et sp. shares characters with both radiodonts and *O. regalis*. We evaluate its phylogenetic position using both maximum parsimony (MP) and Bayesian inference (BI) and further interrogate the support for alternative relationships for *U. comosa* by visualizing the frequency and variation of these alternatives in treespace [34,35]. Treespace visualization provides a comparison of topological incongruence sampled by our analyses, and the distribution of particular clades within those results [34,35]. All analyses support an opabiniid affinity for *U. comosa*. Our results evaluate the uncertainty and relative support for different hypotheses relating to the evolutionary acquisition of characters that define crown group euarthropods.

## 2. Material and methods

### (a) Fossil imaging and measurements
KUMIP 314087, accessioned at the Biodiversity Institute, University of Kansas, Lawrence, Kansas, USA (KUMIP), was photographed using a Canon EOS 500D digital SLR camera and Canon EF-S 60 mm Macro Lens, controlled for remote shooting using EOS Utility 2. Comparative figured material of *Opabinia regalis* is accessioned at the Smithsonian Institution U. S. National Museum of Natural History (USNM). Both polarized and unpolarized lighting were employed, with the fossil surface both wet and dry. Measurements were taken digitally using ImageJ2 [36].

### (b) Morphological matrix
We added five fossil taxa (*Utaurora comosa*, *Amplectobelua symbrachiata* Hou, Bergström & Ahlberg [37], *Houcaris saron* Hou, Bergström & Ahlberg [37], *Cambroraster falcatus* Moysiuk & Caron [18] and *Hurdia triangulata* Walcott [38]) and removed one fossil (Siberian Orsten tardigrade) from a previously published morphological data matrix of panarthropods [39], for a total of 43 fossil and 11 extant taxa. Eighty-six characters were retained from the original matrix, 14 characters were added from two radiodont-focused datasets [16,18], and 25 characters were newly developed or substantially modified herein, for a total of 125 discrete morphological characters. Anatomical features that were only tentatively identified for KUMIP 314087 were coded as '?'. In the case of the proboscis, owing to its uniqueness in *O. regalis* and its relevance to the discussion of the affinity of *U. comosa*, two matrices were generated, one coding this character as present and the other as '?' (further details in electronic supplementary material). Details of all characters including original and new character descriptions and scorings may be downloaded from MorphoBank [40] (www.morphobank.org).

### (c) Phylogenetic analysis
The primary phylogenetic analyses were conducted using BI in MrBayes v.3.2.7 [41], implementing the Markov (Mk) model [42] of character change under two different parameter regimes. We followed the 'maximize information' and 'minimize assumptions' strategies of Bapst *et al.* [43]. The 'maximize information' strategy assumes equal rate distribution across characters and that state frequencies are in equilibrium, as in most previously published BI morphological studies. The 'minimize assumptions' strategy *(a)* applies gamma distributed among-character rate variation, and *(b)* varies the symmetric Dirichlet hyperprior with a uniform distribution of (0,10) to relax assumptions about character state frequency transitions [44]. As with complex molecular substitution models, the 'minimize assumptions' strategy may allow a better fit of the model to the data. Each analysis implemented four runs of four chains each (for 5.5 million and 9.5 million generations, respectively), with 25% burnin. Convergence was assessed based on standard deviations of split frequencies less than 0.01, reaching effective sample size greater than 200 for every parameter, and by comparing posterior distributions in Tracer v.1.7.1 [45].

As the original matrix [39] was devised for MP analysis, we explored MP topologies in TNT v.1.5 [46] using implied weights ($k = 3$) and New Technology. We required the shortest tree to be retrieved 100 times, using tree bisection–reconnection to swap one branch at a time on the trees in memory [47].

### (d) Treespace analysis
Supplemental to traditional clade support metrics, we used classical multidimensional scaling (MDS) to plot treespace [34,35,48,49], with the goal of identifying the distribution of trees resolving key clades formed with *Utaurora comosa* (electronic supplementary material, table S1). Our R script inputs the unrooted post-burnin posterior samples (resultant from BI) and MPTs (resultant from MP) using *ape* v.5.3 [50], and employs *phangorn* v.2.5.5 [51] to calculate pairwise unweighted Robinson–Foulds distances (RF, the proportion of bipartitions defined by a branch in one tree that is lacking in another tree) [52] for the total set of trees resulting from all analyses. The classical MDS function is performed on the RF distances, with a constant added to all elements in the distance matrix

to correct for negative eigenvalues [53]. The treespace therefore approximates the RF distances between trees [34].

# 3. Results

### Systematic palaeontology

Superphylum PANARTHROPODA Nielsen, 1995 [54]
Family OPABINIIDAE Walcott, 1912 [38]

*Diagnosis.* Panarthropod with a short head region bearing a single unjointed appendage (proboscis); slender trunk with dorsally transverse furrows delimiting segments; one pair of lateral flaps per body segment; setal blades cover at least part of anterior margin of lateral flaps; caudal fan composed of multiple pairs of caudal blades; pair of short caudal rami with serrated adaxial margins.

*Type genus. Opabinia* Walcott, 1912 [38].
*Constituent taxa. Utaurora comosa* nov., *Opabinia regalis* Walcott [38].
*Remarks.* See electronic supplementary material.

*Utaurora* nov.

*Etymology.* Concatenation of 'Utah', where the specimen was collected, and 'Aurora', Roman goddess of the dawn who turned her lover into a cicada, reflecting the affinities of this taxon as an early stem group euarthropod.

*Type material, locality, and horizon.* KUMIP 314087, part only, a complete specimen preserved compressed dorso-laterally. Collected by P. Reese from strata of the upper Wheeler Formation (Miaolingian: Drumian), at the Carpoid Quarry (GPS: 39.290417, −113.278519), southwest Antelope Mountain, House Range, Utah, USA [33].

*Diagnosis.* Opabiniid with slender trunk composed of at least 13, likely 15, segments (15 in *Opabinia*); setal structures form blocks that cover the whole width of the trunk and proximal part of the anterior margin of the lateral flaps (setal blades only on flaps in *Opabinia*); tail fan composed of at least seven pairs of elongate and acuminate caudal blades (three pairs in *Opabinia*).

*Utaurora comosa* gen. et sp. nov.

Figures 1*b,c*, 2*b,d*, and 3

2008 *Anomalocaris* sp.: Briggs *et al.*, p. 241, figure 3 [33]
2015 *Anomalocaris* sp.: Robison *et al.*, p. 54–55, fig. 153 (top left) [55]
2021 Incertae sedis: Pates *et al.*, p. 29, table 1. [56]

*Etymology.* 'Comosa' (Latin = 'hairy', 'leafy') reflects the 'hairy' appearance of the dorsal surface, and caudal fan composed of many 'leaves'.
*Diagnosis.* As for genus, by monotypy.
*Description.* KUMIP 314087 represents a complete specimen preserved as a compression in dorsolateral view, with a length (sagittal) of 29 mm (figure 1*b*). The overall organization consists of a short head, an elongate trunk with lateral body flaps and a posterior tail fan. The head and anterior of the trunk are imperfectly preserved; however, fine morphological details can be observed in most of the trunk and tail fan.

The head region measures approximately 10% of the total body length (sag.), and preserves traces of eyes, the mouth and the proboscis. In the ventral posterior region of the head, two curved red structures form an approximately circular outline. This feature could be interpreted as a mouth opening, or alternatively as a poorly preserved eye ('ey? mo?' in figure 2*b*). This possible eye or mouth opening is immediately proximal to a dark red region of one or two

oval shapes, tentatively interpreted as one or two lateral eyes ('ey?' in figure 2*b*). Ventral to this, a cream-coloured elongated conical structure extends from the head ventrally ('pr' in figure 2*b*), with a sub-millimetric orange linear structure of variable width located along its midline ('ic' in figure 2*b*). This is tentatively identified as a proboscis with an internal cavity (figure 2*b*).

The slender trunk (approx. 72% total body length, sag.) is widest towards the anterior and tapers towards the posterior. The dorsal margin bears a 'corrugated' appearance, with indents marking the point where dorsal intersegmental furrows intersect with the margin of the body ('df' in figures 1 and 3). Blocks consisting of dozens of parallel darkly pigmented fine linear structures are arranged along the dorsal furrows and are interpreted as setal blades ('sb' in figures 1 and 3). These blocks extend across the whole width of the trunk and continue laterally over the change in slope on the right side of the body. These setal blocks taper to a rounded subtriangular termination, which overlaps the proximal part of the flaps (figures 1 and 3).

At least 14, likely 15, of these lateral flaps are present on the right side of the body ('fl1?-15' in figure 1). Boundaries are not clear between what are interpreted as the two anteriormost flaps, and these may represent a single flap ('fl1?' in figure 1). Lateral flaps have a subtriangular outline and display a slight taper in size as the body thins posteriorly. The lateral flaps (particularly flaps 6–10) show reverse imbrication with the anterior margin of individual flaps overlapping the posterior margin of the flap immediately anterior to it. The surfaces of the flaps appear smooth and unornamented, with no evidence of strengthening rays or other internal features preserved, but the anterior margins of flaps 2–8 are preserved with a darker coloration compared to the inner region (figures 1 and 3). The posterior flaps (fl13–15) are not completely preserved, especially the ventral margin. A small triangle of setal block present on the anterior margin of the posteriormost flap ('fl15' in figure 1) distinguishes this flap from the caudal blades. However, if instead this setal block is considered to associate with the posterior margin of flap 14, then flap 15 should be treated as part of the caudal fan. Thin structures protruding from beneath flaps 12 and 13 ('?' in figures 1 and 3) are difficult to interpret. They may represent poorly preserved ventral lobopodous limbs, broken margins of swimming flaps, or be artefacts from the matrix.

The posterior of the body (approx. 18% total body length, sag.) consists of a tail fan composed of paired elongate blades, and a pair of caudal rami. The tail has been twisted slightly and the right set of tail blades has been preserved flattened ventrally due to the dorsolateral aspect of preservation. The tail fan has seven, likely eight blades on the left side ('cb' in figure 2), while those on the right cannot be counted with certainty. Unlike the body flaps, these caudal blades are not associated with setal structures. They overlap one another proximally, a given blade largely concealing the blade immediately anterior to it. Each blade has the outline of an elongate parallelogram, longer on the anterior than posterior margin, and their acuminate distal regions splay out. The longest blades measure approximately 4 mm along the anterior margin. Small spines can rarely be seen on the posterior margin of the blade (best seen in figure 3, blade 5). The caudal rami are shorter than any of the caudal blades, measuring approximately 3 mm, and project from

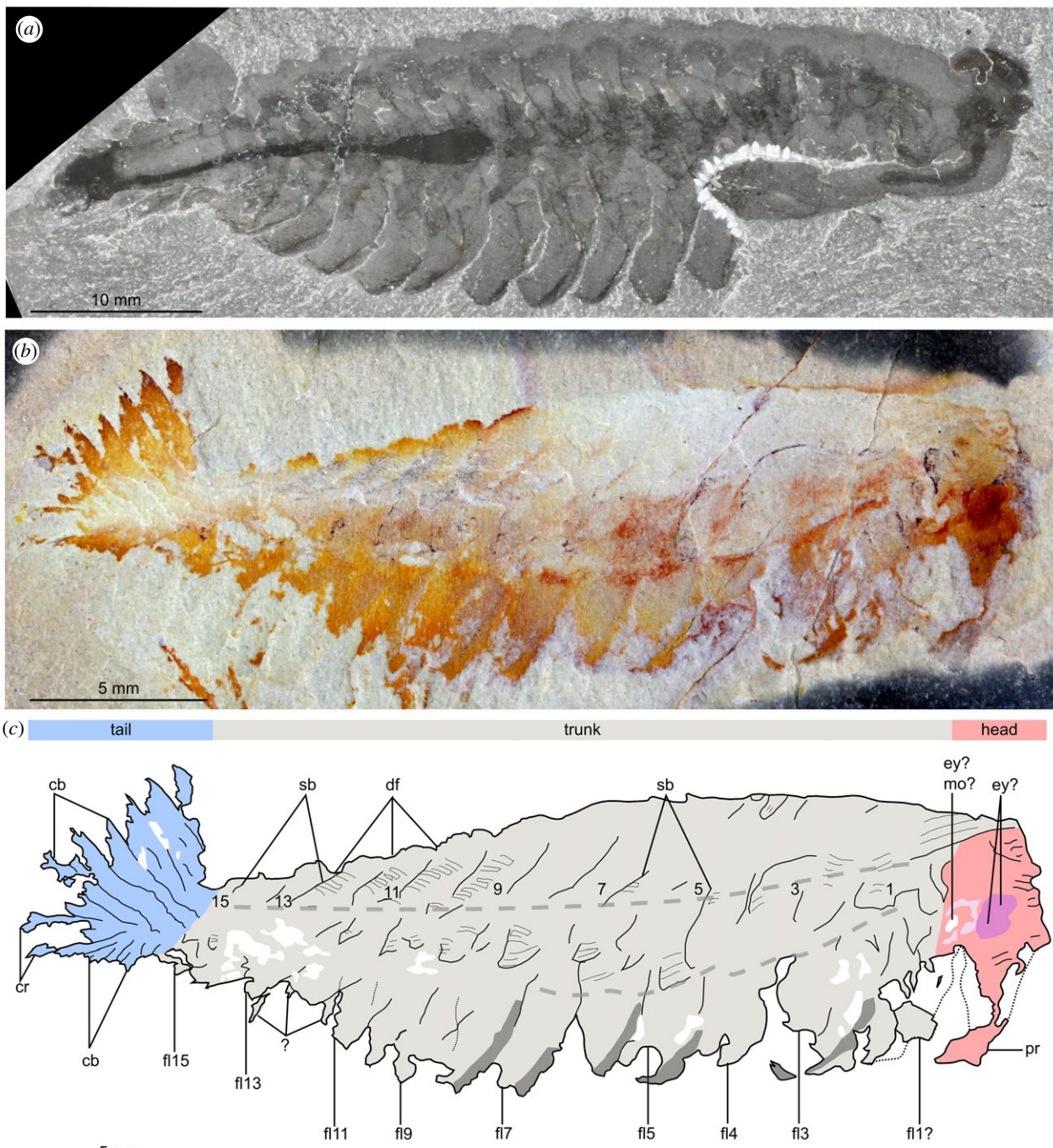

**Figure 1.** Comparison of *Opabinia regalis* Walcott, 1912 from the Burgess Shale (Cambrian: Wuliuan), British Columbia, Canada, and *Utaurora comosa*, gen. et sp. nov., from the Wheeler Formation (Cambrian: Drumian), House Range, Utah, USA. (*a*) USNM 155600, *Opabinia regalis* preserved in lateral view. (*b*) KUMIP 314087, *Utaurora comosa*, preserved in dorsolateral view. (*c*) Interpretative drawing of panel (*b*), dotted lines indicate inferred changes in slope on the body, numbers indicate body segments. Abbreviations: cb, caudal blade; cr, caudal ramus; df, dorsally transverse furrow delineating trunk segments; ey?, dark oval structure in head region, potential eye; fl, lateral flap; ey?mo?, possible eye or mouth; pr, proboscis; sb, setal blade block.

the body at a different angle to them. The two rami appear to diverge from a common point at the posterior of the animal, and exhibit serrated margins ('cr' in figures 1 and 2).

*Remarks.* *Utaurora comosa* was originally described as an anomalocaridid radiodont based on the similarity in the shape of caudal blades to *Anomalocaris* species and the reverse imbrication of the flaps [33]. *U. comosa* also shares with some radiodonts the presence of setal blades that extend over the dorsal midline of the body. The recognition herein of a putative proboscis with internal cavity, dorsally transverse furrows that delimit segments in the trunk, and a pair of short caudal rami with serrated axial margins, support closer affinities of this animal with *Opabinia regalis*,

rather than with *Anomalocaris*. The unique combination of characters, and novel features such as the elaborate tail fan, warrant the erection of a new genus and species.

Among members of the euarthropod lower stem group, a proboscis has only been reported previously in *Opabinia* [7]. The proboscis of *U. comosa* protrudes from the head in a similar position relative to the tentatively interpreted eyes as in *Opabinia*. In addition, a feature comparable to the internal cavity within the proboscis of *Opabinia* can be observed in *Utaurora* (figure 2). However, unlike *Opabinia*, no annulations can be seen in this structure, as it is too poorly preserved. *U. comosa* also has dorsal furrows delineating the body segments. Such dorsal epidermal segmentation is seen in *Opabinia* but

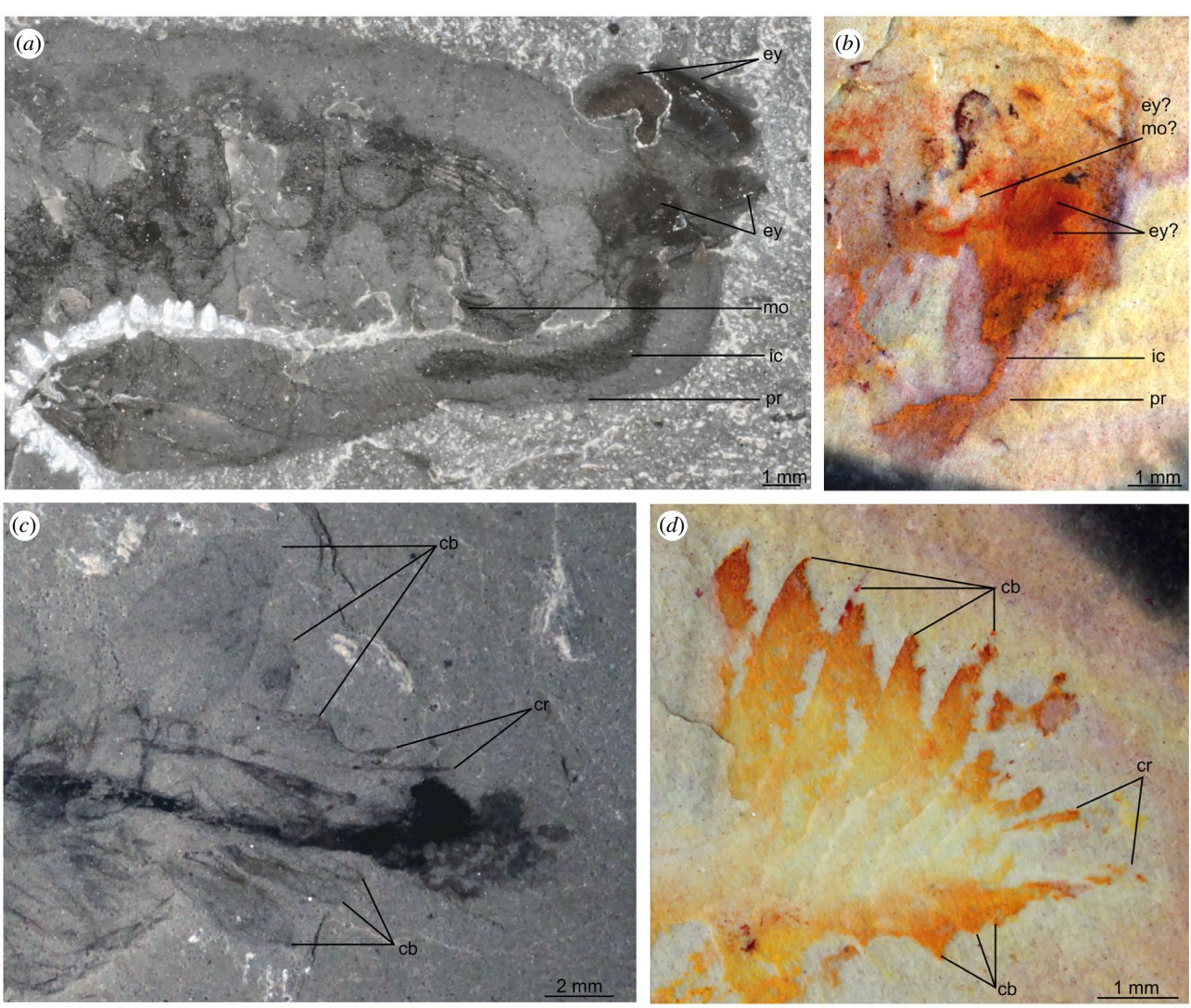

**Figure 2.** Details of the head and tail regions of *Opabinia regalis* Walcott, 1912 (*a,c*) and *Utaurora comosa* gen. et sp. nov. (*b,d*). (*a*) Head region of USNM 155600, *Opabinia regalis*, showing eyes, posterior-facing mouth, and proboscis with internal cavity. (*b*) Head region of KUMIP 314087, *Utaurora comosa*, showing possible eyes, mouth and putative proboscis with internal cavity. (*c*) Tail region of USNM 155600, *Opabinia regalis*, showing lobate tail blades, paired caudal rami with serrated adaxial margin, and posterior body termination extending beyond posteriormost caudal blades and caudal rami. (*d*) Tail region of KUMIP 314087, *Utaurora comosa*, (photo mirrored), showing caudal blades and caudal rami with serrated adaxial margin. Abbreviations: cb, caudal blade; cr, caudal ramus; ey, eye; ic, internal cavity of proboscis; mo, mouth; pr, proboscis.

is unknown in all other lower stem group euarthropods (including *Kerygmachela*, *Pambdelurion* and all radiodonts) [27].

*Utaurora comosa* also displays characters known in both radiodonts and *O. regalis*. The slender, broadly rectangular dorsal outline of the body in *U. comosa* is comparable to what is observed in both *O. regalis* and the radiodonts *Aegirocassis benmoulae* and *Hurdia* spp. This outline contrasts with the diamond-like outline of many radiodonts, including *Amplectobelua symbrachiata*, *Anomalocaris canadensis* and *Peytoia nathorsti* [8,10,57]. In addition, both *O. regalis* and radiodonts possess setal blades, in varying arrangements (electronic supplementary material, figure S1). In *A. benmouale* and *P. nathorsti*, these structures form a single block per body segment, which covers the width of the trunk [14], while in *O. regalis* the setal structures cover the anterior margin of the flaps [9]. *U. comosa* appears to display a combination of these two states, with setal blades covering the dorsal surface in a single block, which extends laterally to the proximal region of the anterior margins of corresponding flaps (electronic supplementary material, figure S3). Strengthened anterior margins of lateral flaps have also been reported in a juvenile specimen of

the amplectobeluid radiodont *Lyrarapax unguispinus* [17], and can also be observed in *O. regalis*, where they are preserved with a distinct elemental signature [32]. A tail fan associated with caudal rami is also known in both *O. regalis* and some radiodonts, though the number of blades known in *U. comosa* (at least seven, likely eight, on each side) by far exceeds what is known in either *O. regalis* (three) or any radiodont (ranging from zero to three). The acuminate tips of elongate caudal blades of *U. comosa* are most similar in morphology to those of *An. canadensis*, and contrast to the more lobate caudal structures known in *O. regalis* and other radiodonts such as *Hurdia* (figure 2) [7,10,57,58], however spines are only known on the caudal blades of *O. regalis* (electronic supplementary material, figure S2). Paired caudal rami have been reported in *Am. symbrachiata*, *Houcaris saron* (formerly *An.* [59]) and *L. unguispinus* [12,57], though these are much more elongate than in both *U. comosa* and *O. regalis* and lack the serrated margin common to the opabiniid taxa (figure 2 and electronic supplementary material, figure S2) [7,57]. The body is prolonged posteriorly by an unpaired and non-serrated structure in *An. canadensis*, *Kerygmachela kierkegaardi*, and

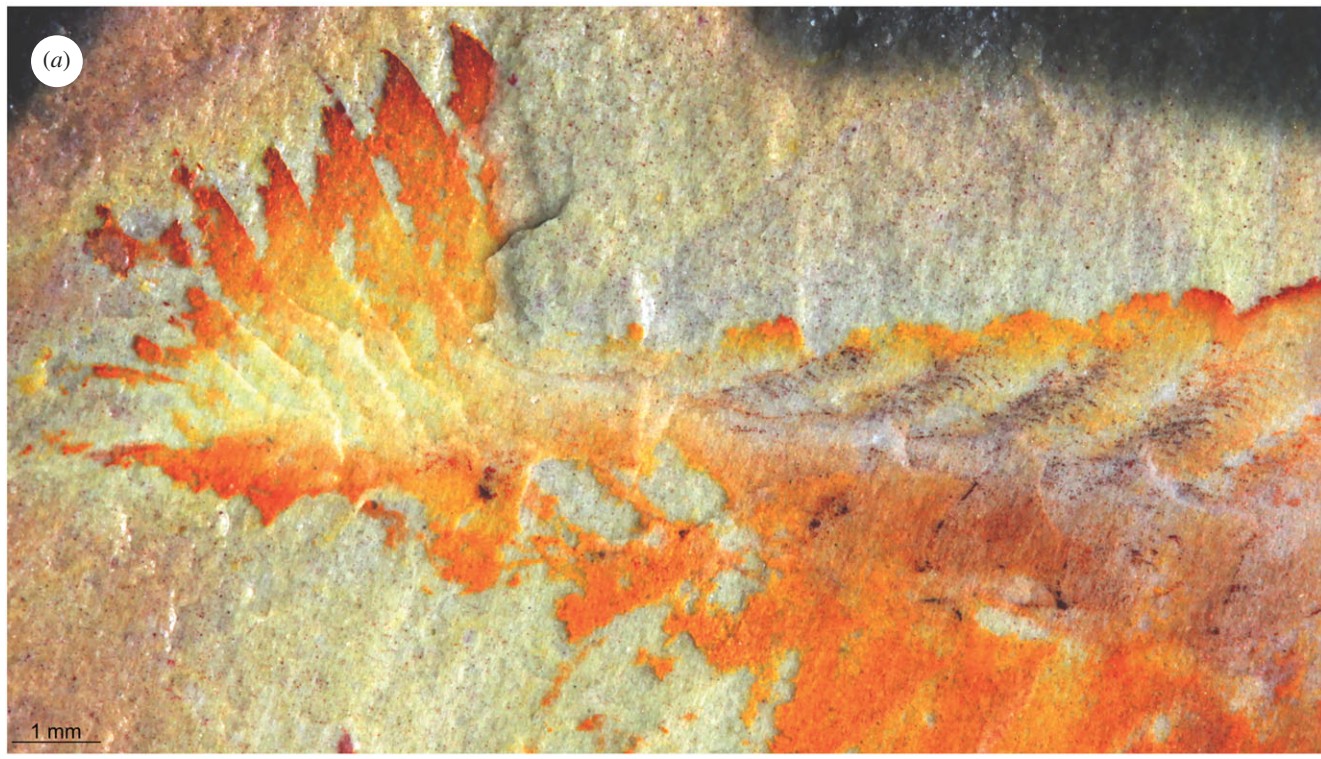

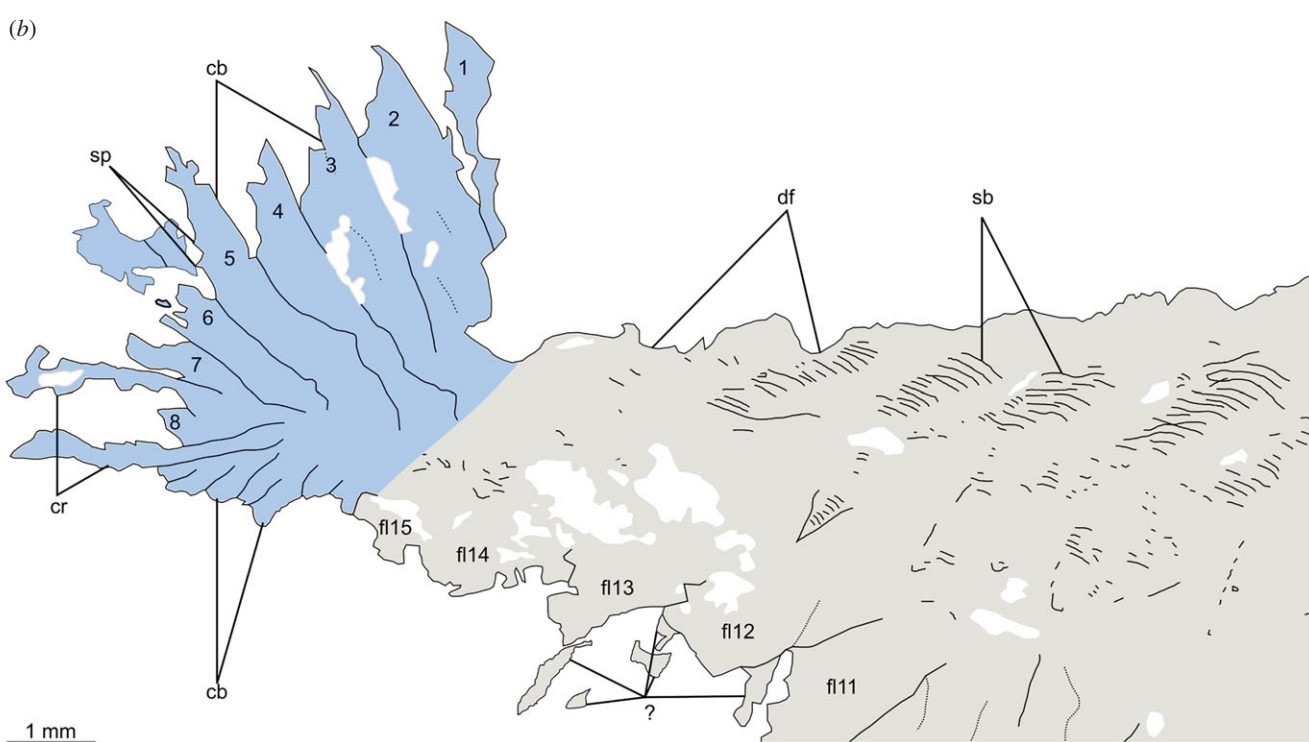

**Figure 3.** Posterior of KUMIP 314087, *Utaurora comosa* gen. et sp. nov. including details of setal blade blocks, elaborate caudal fan and paired caudal rami. (*a*) Photograph. (*b*) Interpretative drawing. Abbreviations: cb, caudal blade; cr, caudal ramus; df, dorsally transverse furrow delineating trunk segments; fl, lateral flap; sb, setal blade block; sp, spine on caudal blade.

*Schinderhannes bartelsi* (e.g. [10,19,31]), which may represent fused caudal rami or alternatively, a non-appendicular tail spine. Regardless, all these unpaired terminal structures are much longer relative to the main body than the caudal rami of opabiniids, and none exhibit serrated margins.

*Phylogenetic results*. To test the affinities of *Utaurora comosa* relative to *Opabinia regalis* and radiodonts, we scored this specimen into a morphological matrix. Regardless of whether the matrix was analysed with BI (figure 4*a* and electronic supplementary material, figure S3a,b) or MP (electronic supplementary material, figure S3c), a clade comprising *U. comosa* and *O. regalis* was resolved, warranting the assignment of the new taxon to family Opabiniidae. As the evidence for a proboscis in *U. comosa* is tentative (figure 2*b*), we conducted sensitivity analyses by building phylogenies where the proboscis (character 14) was coded as uncertain. With BI, opabiniids remained monophyletic (with lower nodal support; electronic supplementary material, figure S4a,b). With MP and an uncertain proboscis, the monophyly of opabiniids collapsed to a polytomy with deuteropods (electronic supplementary material, figure S4).

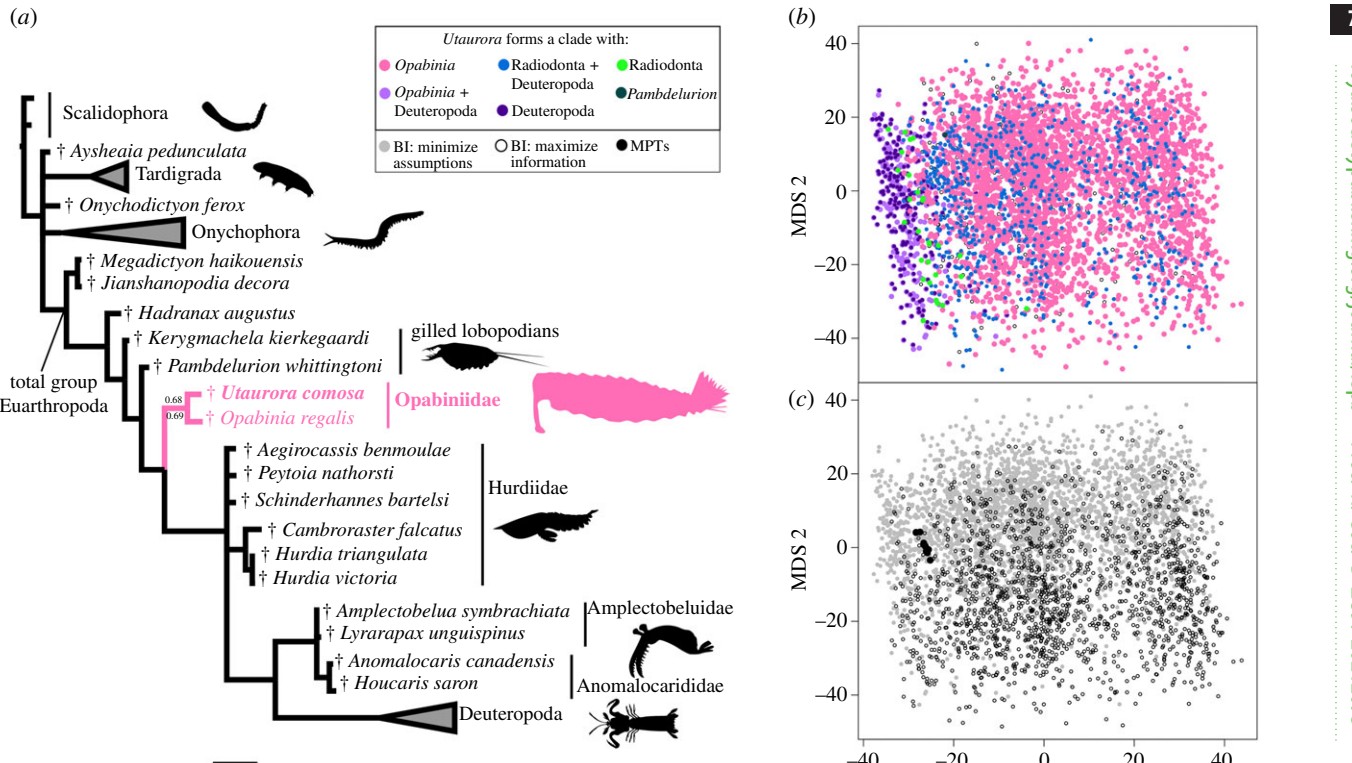

**Figure 4.** Phylogenetic relationships of opabiniids and lower stem group euarthropods. (a) Summarized topology based on the consensus tree retrieved with BI under minimize assumptions parameters. Numbers at the key node indicate posterior probabilities from this analysis, and from BI under maximize information parameters. Credits for silhouettes: *Priapulus caudatus*: Bruno C. Vellutini (CC BY 3.0); *Hurdia victoria* and *Anomalocaris canadensis*: Caleb M Brown (CC BY-SA 3.0) (b) Treespace plotted by bipartition resolving *Utaurora*. Points are coloured by relationships for this taxon. (c) Treespace plotted by analysis.

As the support values were poor for a morphological analysis (figure 4a and electronic supplementary material, figure S3: posterior probabilities of 0.68 and 0.69 with BI; jackknife value of 57 and GC value of 65 with MP), we visualized treespace [34]. Such methods may be especially useful for fossils with a greater degree of uncertainty in their interpretation, as with *U. comosa*. Our plots identify whether uncertainty in support for opabiniid relationships in the posterior sample ($n = 4512$ trees for analyses where proboscis is coded as present; electronic supplementary material, table S1) and MPTs ($n = 12$ trees) is restricted to tree islands with otherwise similar topologies, or spread throughout a large region of occupied treespace. While treespace has been previously explored in meta-analyses of fossil datasets [35,60,61], this is, to our knowledge, the first attempt to use such a visualization to interrogate the distribution of bipartitions for the position of a focal fossil taxon. Several possible hypotheses are subsets: *Utaurora* could be part of a clade with either *Opabinia* or Deuteropoda (pink and dark purple colours, respectively, in figure 4b), and could be part of both those clades (light purple in figure 4b). Our overall treespace for *Utaurora* can nevertheless be grouped by islands of trees where the supermajority of trees are related to opabiniids ($n = 3102$ trees total for analyses where proboscis is coded as present) or a minority to deuteropods ($n = 251$ trees total). A sparse, slender zone ($n = 28$ trees total) of the alternative exclusive hypothesis that *Utaurora* is a radiodont [33] transitions between the opabiniid and deuteropod islands. Interspersed sparsely within the opabiniid island are topologies supporting *Utaurora* with both radiodonts and deuteropods, but excluding *Opabinia* (blue in figure 4b); most of these trees depict *Opabinia* as the direct outgroup rather than a wildcard taxon (occupying different positions that are topologically

distant). Choice of BI model parameters did not substantially impact the treespace (figure 4c: grey and open circles overlap completely on axis 1 and much of axis 2), while the MPTs (figure 4c: black circles) formed a small but distinct cluster.

## 4. Discussion

### (a) The power of treespace for phylogenetic uncertainty of fossils

At first glance, our phylogenetic analyses provide only weak nodal support for the placement of *Utaurora comosa* within Opabiniidae. Although similar nodal support with a similar data matrix has been used to reclassify enigmatic fossils [62], we further interrogated our results—especially important as our terminal of interest is represented by a single specimen with some characters that are difficult to interpret. Therefore, we investigated the degree of uncertainty among contributing bipartitions, finding an increased number of topologies (electronic supplementary material, table S1) that support *U. comosa* forming a clade with *Opabinia regalis*, and not with an alternative taxon. Such calculations have been effective in summarizing the taxonomic uncertainty in fossil placement [63]. Furthermore, our visualization of the sample of optimal trees [34,35,49] illustrates the distribution of topological distances between conflicting and overlapping hypotheses, whether these form separate tree islands (e.g. alternate positions of *U. comosa* in figure 4b) or are broadly distributed throughout the entire topological space (e.g. support for some radiodont clades in electronic supplementary material, figure S5c). This technique allows the strength of support

for competing hypotheses of relationships to be more comprehensively evaluated beyond an arbitrary cut-off value.

Phylogenetic analyses aiming to resolve the relationships of fossil taxa present challenges such as researcher-specific morphological interpretation and coding decisions, preponderance of missing data (common for exceptionally preserved Cambrian taxa, due to preservation of few specimens or taphonomic loss of labile morphology), and relatively simple models of character change that may not reflect true evolutionary history [64–68]. Visualization of treespace investigates how these scenarios may affect a consensus topology. In the case of *U. comosa*, the morphological description is based on a single specimen where we could only tentatively identify the proboscis. Therefore, we compared alternative codings to represent our uncertainty in interpretation, and the potential influence on the definition of opabiniids (electronic supplementary material, figures S4 and S6a,b). The sister group relationship of *U. comosa* with *O. regalis* (rather than radiodonts or deuteropods) is not driven solely by the proboscis character, and is maintained due to a suite of shared morphological characters (e.g. dorsal furrows, caudal rami and proboscis).

## (b) Implications for opabiniid evolution and ecology

Our phylogenetic results provide substantial support for an assignment of *Utaurora comosa* to Opabiniidae, helping to clarify some debates about the morphology of *Opabinia regalis*. Two contrasting interpretations have been presented for the relationship between the lateral flaps and the blocks of setal blades in *O. regalis*: one where the setal blades are attached to the dorsal surface of the lateral flaps [9,23], and the other view suggesting the setal blades were attached as a fringe along the posterior margin of the lateral flap [32]. The setal blades in *U. comosa* support the former interpretation, with the setal blades extending mainly along the dorsal surface of the body but also along the basal anterior margin of the flaps.

The family Opabiniidae is now considered to comprise two taxa, expanding its range geographically from two quarries separated by only a few metres to two deposits approximately 1000 km apart during two Cambrian stages [69]. Although both *O. regalis* and *Anomalocaris canadensis* underwent major redescriptions around the same time [7,8,70], our revised opabiniids have not nearly caught up to the known diversity or distribution of radiodonts (or even the monophyletic groupings recovered in this study, Hurdiidae and Amplectobeluidae + Anomalocarididae). Radiodont frontal appendages, mouthparts, and carapaces are sclerotized and are often among the first fossils recovered from Cambrian deposits preserving non-biomineralizing organisms, and indeed many radiodont taxa are only known from their frontal appendages (e.g. [20,71]). However, preservation potential alone is insufficient to account for the greater diversity and distribution of radiodonts relative to opabiniids, as even radiodonts known only from complete specimens greatly outnumber opabiniids, both globally and within the Burgess Shale. Thus, the absence of opabiniids in other deposits from which complete radiodonts are known likely reflects a true absence or much lower diversity.

## (c) Implications for the euarthropod stem group

Our results have implications for larger scale questions, such as the relative phylogenetic positions of opabiniids and radiodonts along the euarthropod stem group, and detailed consideration

of conflicting topologies. We replicate the dichotomy of recent publications, where matrices analysed using MP find opabiniids as the sister group to deuteropods [39,62] and those analysed using BI or maximum-likelihood instead resolve radiodonts in that position [18,62,72,73]. The branching order of these three clades has ramifications for the sequence of acquisition, and evolutionary reversals or convergences, of key crown group euarthropod characters [27], such as the posterior mouth and arthropodized appendages, as well as the dorsal expression of trunk segmentation (electronic supplementary material, figure S6). The scenario (favoured by MP and an island of BI topologies) where opabiniids are sister group to deuteropods requires either the secondary loss of arthropodized appendages in opabiniids, or the convergent evolution of arthropodized appendages in radiodonts and deuteropods.

The consensus topology (figure 4a and electronic supplementary material, figure S6a), and the majority of topologies (yellow, pink and maroon points in electronic supplementary material, figure S6c), support a single origin of arthropodization in euarthropods. A possible developmental framework would entail the single anterior protocerebral pair of arthropodized limbs in radiodonts becoming co-opted posteriorly to enable the arthropodization of all limbs [74,75]. This scenario would require the convergent fusion of presumed protocerebral appendages in opabiniids to form a single proboscis, and of protocerebral limb buds in deuteropods to form the labrum [15,31,74,76]. Evolutionary reversals or convergences are also required by these topologies (electronic supplementary material, figures S6 and S7). The posterior-facing mouth shared by *Opabinia regalis* and deuteropods is either convergent or lost in radiodonts [15]. Additionally, the distinct dorsally transverse furrows delineating segment boundaries (reported in both opabiniids), which may represent a precursor to arthrodized tergites in deuteropods [77], could either be lost in radiodonts and regained in deuteropods, or represent a convergent expression of dorsal trunk segmentation.

The consensus topology is further complicated by the apparent paraphyly of radiodonts (figure 4a and electronic supplementary material, figures S3a,b and S4b). Traditional nodal support resolves a clade of amplectobeluids, anomalocaridids and deuteropods with posterior probabilities of 0.52–0.61 (electronic supplementary material, figures S3a,b, S4a,b). The specific relationship of amplectobeluids and anomalocaridids with deuteropods might improve some aspects of limb evolution, as the loss of dorsal flaps (shared by opabiniids and hurdiids; electronic supplementary material, figure S1) prior to the proposed fusion of setal blades and ventral flaps into the deuteropod biramous limb removes the requirement to identify a dorsal flap homologue in deuteropods [14]. However, treespace visualization does not provide strong support for radiodont paraphyly, as overlapping islands resolve conflicting relationships among radiodonts and deuteropods (electronic supplementary material, figures S5c and S6c and discussion). As many of the characters distinguishing internal relationships among radiodont families describe the protocerebral frontal appendages, and are coded as inapplicable to all other taxa, we propose revised models of character evolution [66,67] may be necessary to resolve these relationships; accordingly we place little weight on this particular result. It should be emphasized, however, that the position of *Utaurora comosa* is not affected by this uncertainty, as its position as sister taxon to each radiodont clade was tested (with only non-zero results reported in (electronic supplementary material, table S1).

# 5. Conclusion

The 'weird wonders', as popularized by Gould [3], inspired a generation of Cambrian palaeontologists, with *Opabinia regalis* at the heart of his narrative. The reorganization of previously enigmatic Cambrian taxa into stem groups instead revealed their importance for reconstructing the origins of modern phyla. Resolving the phylogenetic placement of these species is crucial for understanding the sequence of evolution of diagnostic crown group characters, as well as reconstructing the diversity and palaeogeography of early ecosystems and groups. Here we apply treespace visualization to the reinterpretation of the relatively poorly preserved fossil *Utaurora comosa*. Dissection of the phylogenetic support demonstrates that while evidence for radiodont paraphyly is weak, *U. comosa* can be confidently reassigned to Opabiniidae. The weirdest wonder of the Cambrian no longer stands alone.

Data accessibility. Electronic supplementary data files are available at MorphoBank (www.morphobank.org) doi:10.7934/P3874 and from the Dryad Digital Repository: https://doi.org/10.5061/dryad.6t1g1jwz4 [78].

Nomenclatural acts relating to the new taxon are registered on ZooBank. LSID urn:lsid:zoobank.org:pub:E44CEC6A-4C32-42EC-BA64-19D37FD58665 (publication); LSID urn:lsid:zoobank.org:act:CAA92ACF-2CAB-45B4-96E3-38B34A2974F9 (genus); LSID urn: sid:zoobank.org:act:5B8B3E8F-A339-4A28-A5EB-196C4E099A7B (species).

Authors' contributions. S.P. and J.M.W.: conceptualization, data curation, formal analysis, funding acquisition, investigation, methodology, project administration, resources, software, supervision, validation, visualization, writing—original draft, writing—review and editing; A.C.D.: investigation, resources, supervision, writing—review and editing; R.L.-A.: investigation, writing—review and editing; J.O.-H.: funding acquisition, investigation, resources, supervision, writing—review and editing.

All authors gave final approval for publication and agreed to be held accountable for the work performed therein.

Competing interests. The authors declare that they have no competing interests.

Funding. S.P. acknowledges funding from an Alexander Agassiz Postdoctoral Fellowship (Museum of Comparative Zoology, Harvard University) and a Herchel Smith Postdoctoral Fellowship (University of Cambridge). This work was also supported by the National Science Foundation DEB #1856679 to J.M.W. and J.O.-H.

Acknowledgements. We would like to acknowledge the Goshute, Southern Paiute and Ute tribes, on whose lands this fossil specimen was collected. We thank three reviewers and the associate editor for their comments on the manuscript. We are grateful to P. Reese, who collected KUMIP 314087 and generously donated it to the Biodiversity Institute, University of Kansas (KUMIP). Access to and loan of this specimen was facilitated by B.S. Lieberman and J. Kimmig (KUMIP). J.O.-H. thanks S. Whittaker (Smithsonian Institution) for facilitating access to and training for the imaging facilities. We thank M.J. Hopkins (American Museum of Natural History) and L.T. Rangel (Massachusetts Institute of Technology) for discussions about treespace, and F. Anthony for collaboration on the fossil reconstruction.

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
