## [Peer Review File · Proceedings of the Royal Society B: Biological Sciences]

Review History

RSPB-2021-2093.R0 (Original submission)

Review form: Reviewer 1

Recommendation

Accept with minor revision (please list in comments)

Scientific importance: Is the manuscript an original and important contribution to its field?

Excellent

General interest: Is the paper of sufficient general interest?

Good

Quality of the paper: Is the overall quality of the paper suitable?

Good

Is the length of the paper justified?

Yes

Should the paper be seen by a specialist statistical reviewer?

No

Do you have any concerns about statistical analyses in this paper? If so, please specify them explicitly in your report.

No

It is a condition of publication that authors make their supporting data, code and materials available - either as supplementary material or hosted in an external repository. Please rate, if applicable, the supporting data on the following criteria.

Is it accessible?

Yes

Is it clear?

No

Is it adequate?

Yes

Do you have any ethical concerns with this paper?

No

Comments to the Author

Comments to authors

The iconic opabinid is among the most hotly debated animals from the Cambrian. Only one species, *Opabinia regalis* from the Burgess Shale, has been known for over one century. Here Pates and colleagues reinterpret an old specimen from the Wheeler Formation of Utah as the second representative of opabiniids. The authors further conduct phylogenetic analysis with treespace visualization, revealing that the new named animal being a member of opabiniids. I find this work interesting in general and I agree that this specimen might be opabinid. But several interpretations in the manuscript, such as mouth opening, extension of setal blades and the number of caudal blades, might need a more cautious manner though.

1) the mouth

The authors interpreted a patch of exposed matrix right behind the eyes, at best with a darker circle outside, as potential mouth of the new opabiniid (Fig 2b). While in *Opabinia regalis*, the mouth is located further back beneath the first trunk segment, primarily due to the backward folding of the esophagus (Whittington 1985). With this ambiguity in structure and difference in position, I hesitate to believe the mouth is preserved.

2) the setal blades

The authors argue that the state of setal blades in Utah opabiniid represent a combination of some radiodonts and *Opabinia* (lines 232-237), they cover the whole dorsal surface and extend to the anterior margin of flap proximal part. I am not so sure about the relative position of setal blades and the flaps. The specimen and interpretative drawing (Fig 1b,c) show that setal blades can extend to the anterior (flaps 10, 11), middle (flaps 5, 6) or even rear margin (flap 9?) of the corresponding flap proximal part. This variation might indicate that the setal blade of Utah opabiniid is more radiodont-like, it does not attach with the flaps.

3) the lobopodous limbs

The interpretation of lobopodous limbs is already quite radical. These slender reddish patches with irregular margin are not well distinct from the flaps (normally broken with irregular margin). It really drives me into crazy when I saw the indicative segment/annulation lines in the drawings (Fig 1c, 3b), particularly by the one in fig 3b.

4) the number of caudal blades

The author interpreted that Utah opabiniid have at least seven, probably 8 pairs of caudal blades, a much, much larger number in radiodonts and Opabinia. I am not convinced by this interpretation. There is an alternative possibility. When you trace the trunk (marked by seta), you can see that the body axis (which should divide the pairs of the caudal blades) go to the fourth blade counting from the left side. Given the dorsal-lateral compression of the specimen and the overlapping the caudal blades, the one that is most close to us should be exposed as the largest, which should be edged by the two 'caudal ramus'.

In addition, it is always radical to erect new taxa based on single poorly preserved specimen. The sole specimen of Utah opabiniid is nearly complete. However, with the absence of features of proboscis, the only character that differs Utah specimen from Opabinia regalis is the number of caudal blades. It would not be so well-grounded to do taxonomic exercise here.

I would recommend to publish this manuscript after a minor revision.

Review form: Reviewer 2

Recommendation

Accept with minor revision (please list in comments)

Scientific importance: Is the manuscript an original and important contribution to its field?

Excellent

General interest: Is the paper of sufficient general interest?

Excellent

Quality of the paper: Is the overall quality of the paper suitable?

Excellent

Is the length of the paper justified?

Yes

Should the paper be seen by a specialist statistical reviewer?

No

Do you have any concerns about statistical analyses in this paper? If so, please specify them explicitly in your report.

No

It is a condition of publication that authors make their supporting data, code and materials available - either as supplementary material or hosted in an external repository. Please rate, if applicable, the supporting data on the following criteria.

Is it accessible?

Yes

Is it clear?

Yes

Is it adequate?

Yes

Do you have any ethical concerns with this paper?

No

Comments to the Author

This paper describes a new fossil opabiniid, a stem-group euarthropod from the Cambrian period. The new taxon is known from a single specimen, and the quality of preservation (while exceptional) leaves some characters in doubt. However, the rarity of opabiniids in the fossil record (this is only the second known species), combined with their pivotal position in the arthropod tree (close to the emergence of the various key characters), amply justifies publication, in my view. The authors have been very honest about the limitations of the fossil material, and take a very measured approach to considering its significance. Indeed, the value of the paper lies as much in its careful methodological approach as in the novelty of the fossil material, and is further enhanced by a detailed exploration of character evolution in this critical part of the arthropod tree. Therefore, I expect the paper to have broad appeal, far beyond specialists in stem-group euarthropods, and the methods could set a new standard for dealing with problematic fossils, and a new way of visualizing support for phylogenetic hypotheses. I recommend the paper for publication with only very minor changes.

Detailed comments:

Placement of new taxon: in general, the worry with a single specimen with partly ambiguous morphology is that a small number of homology assessments can essentially channel the outcome of any phylogenetic analysis. I'm relieved that the authors have faced this potential pitfall head-on by running sensitivity analyses in which the canonical opabiniid character - the 'proboscis' - is coded as uncertain. However, I wonder how sensitive the opabiniid grouping is to a small number of other characters, e.g. the caudal rami (which I find perhaps least convincing of all the described characters). I'm not suggesting that the authors run new sensitivity analyses, but perhaps they could comment on whether they expect this to be a problem. To put it another way, why test the coding of the proboscis, but not other potentially decisive characters?

Abstract and set-up of the radiodont vs. opabiniid comparison: in the abstract (p. 1, lines 20-22) it seems strange to make the point that *Anomalocaris* and relatives are very diverse but *Opabinia* (till now) is a one-off. This idea is picked up again in the discussion (p. 18, lines 355-356): "our revised opabiniids have not nearly caught up to the known diversity or distribution of radiodonts". Why would one compare groups like this? It doesn't seem to be meaningful, especially if radiodonts are paraphyletic. I'd be inclined in the abstract to make the point that every new species from this part of the tree has the potential to usefully constrain arthropod character evolution, especially a new representative of a poorly known group. In the discussion (lines 371-374), I find the suggestions of an ecological explanation for relative diversity of radiodonts and opabiniids rather speculative (and again, a bit meaningless if radiodonts are not a clade). It seems unfair to say that opabiniids show limited evidence for adaptations for different niches, when there's only two of them!

Figures and described anatomy: I'm fascinated by the 'setal blocks' and the inclusion of a magnified image in Fig. 3 is welcome. Do you have even more magnified images that might be included in the supplementary info? (I appreciate the specimen is very small). The reverse imbrication of flaps is a potentially pivotal character, but I don't feel confident that I can see it in the images, and especially not in the interpretative line drawings. Please consider a detailed photo, or a tweak to the line drawing to make this clear.

Minor comments:

Phrase 'hotly contested/hotly debated' sounds rather journalistic - but elsewhere I appreciate that you've tried to explain exactly how the competing hypotheses arise

Lines 79-83: unclear phrasing in “The identification of...”: are you talking in general now about what’s needed to distinguish plesiomorphies and apomorphies, or are you referring to Opabinia in particular?

Line 155 and elsewhere: I find the description of setal blocks extending across the “entire dorsal surface” a bit confusing: you mean across the whole width of the ‘trunk’ and also the flaps (but “entire” suggests they’re also covering the tail and head - ?)

Line 156: “These setal structures...” - you mean the setal blocks taper laterally to a rounded termination? (first I read it as meaning the ‘setae’ have rounded tips)

Line 266: you refer here and in sup fig 4 to KUMIP 314087, but elsewhere use the new taxon name - is this a deliberate distinction?

Line 317: “related to at least one opabiniid” - but there’s only one (Opabinia)?

Supp figures 3 and 4 - the trees are too small/low-res for me to read some of the text (especially the support metrics). This may just be in my review copy but please check the final submitted versions are clear.

Review form: Reviewer 3

Recommendation

Accept with minor revision (please list in comments)

Scientific importance: Is the manuscript an original and important contribution to its field?

Acceptable

General interest: Is the paper of sufficient general interest?

Good

Quality of the paper: Is the overall quality of the paper suitable?

Good

Is the length of the paper justified?

Yes

Should the paper be seen by a specialist statistical reviewer?

Yes

Do you have any concerns about statistical analyses in this paper? If so, please specify them explicitly in your report.

No

It is a condition of publication that authors make their supporting data, code and materials available - either as supplementary material or hosted in an external repository. Please rate, if applicable, the supporting data on the following criteria.

Is it accessible?

Yes

Is it clear?

Yes

Is it adequate?

Yes

Do you have any ethical concerns with this paper?

No

Comments to the Author

This manuscript redescribed a fossil specimen KUMIP 314087 from the Wheeler Formation of Utah, which was originally described as an anomalocaridid radiodont. In this manuscript, the authors carried out a careful comparative morphological study and novel phylogenetic and treespace visualization analyses, which suggest that this animal has a close affinity with the Burgess Shale animal *Opabinia*, contributing to our understanding of the diversity and evolution of opabiniids. The manuscript was very well written, the description and analyses are generally robust, and the conclusion and discussion are sound. Therefore, I would like to support its publication on Proceedings B after some minor revisions.

- The structure of the manuscript

1) There are two key novel points in this manuscript: the discovery of an opabiniid from Utah and the novel analytical method of treespace visualization. However, I felt that these two parts are slightly disjointed and lack a good coherency throughout the manuscript. Maybe the authors can emphasize the phylogenetic uncertainty of KUMIP 314087, and then how to use this new method to solve the problem.

2) It is better to have the Material and Method section after the Introduction, so the readers can understand the results better.

- Morphological details

1) It is better to add some explanation of the specimen preservation in the first paragraph of the description before going into morphological details. As the specimen is poorly preserved, some morphological interpretations are not very convincing.

2) Line 142-143: “two curved red structures surround a circular opening, interpreted as a mouth opening”. – This structure is not well defined, and the interpretation is not very convincing. It is very close to the possible eyes, and they are all in a similar shape and size. Is it possibly being an eye but preserved slightly different due to missing the surface? By comparison with Burgess Shale *Opabinia*, the mouth position should be much more ventral and posterior. This structure seems to be too close to the proboscis, which would be very difficult for feeding.

3) Line 144: “interpreted as a pair of lateral eyes” – By comparison with *Opabinia*, I agree that this structure seems to be eye(s), but it is difficult to tell if paired or not.

4) Line 159: “At least 14, likely 15” – I feel that the flaps 11-15 are not completely preserved and miss some parts toward their ventral side margin. The current “flap 14/15” should correspond to the caudal blade 1/2 on the right side.

5) Line 160: “Boundaries are not clear between what are interpreted as the two anteriormost flaps,” – It is true that the boundaries are not clear, but they seem to be clearer in Fig. 1b than in Fig. 1c. The camera-lucida drawing doesn't match well with the specimen in some details. I wonder if a low-angle light will help to solve the boundaries of these flaps.

6) I barely can see the structures interpreted as lobopodous limbs, which is the least convincing structure in the description.

Overall, I agree that *Utaurora* shows many characters like *Opabinia*, and I have no problem interpreting the proboscis, setal blades, flaps, caudal blades, and caudal rami. Therefore, there is no need to overinterpret some uncertain structures.

- Phylogenetic analyses

1) Please check how the uncertainty of these characters above might influence the phylogenetic results.

2) *Kylinxia* was recently reported from the Chengjiang biota (Zeng et al. 2020), and it shows similar eye morphology with *Opabinia*, and it is also suggested to be at an important phylogenetic position, so please add some comparative discussion with *Kylinxia*, and this taxon should also be included in the phylogenetic analyses.

- Treespace visualization

This is a relatively new method and “the first attempt to use such a visualization to interrogate the distribution of bipartitions for the position of a focal fossil taxon”. Although the authors cited some previous references, it will still be helpful to summarise the methods and principles of this method and explain its advantage compared to traditional methods. Otherwise, I feel that Table 1 is very clear and convincing, and the visualisation in Fig. 4b/c seems unnecessary.

- Others:

Line 55: “the most” change to “one of the most”

Line 79: “remains hotly debated.” – please add relevant references here

Line 121: “of at least 13, likely 15, segments”, in order to be consistent with sentences below, it is better to add “(xx segments in Opabinia)”.

Line 138: “(Fig. 1)” change to “(Fig. 1b)”

Decision letter (RSPB-2021-2093.R0)

22-Nov-2021

Dear Dr Pates:

Your manuscript has now been peer reviewed and the reviews have been assessed by an Associate Editor. The reviewers’ comments (not including confidential comments to the Editor) and the comments from the Associate Editor are included at the end of this email for your reference. As you will see, the reviewers and the Editors have raised some concerns with your manuscript and we would like to invite you to revise your manuscript to address them.

When submitting your revision please upload a file under "Response to Referees" - in the "File Upload" section. This should document, point by point, how you have responded to the reviewers’ and Editors’ comments, and the adjustments you have made to the manuscript. We require a copy of the manuscript with revisions made since the previous version marked as ‘tracked changes’ to be included in the ‘response to referees’ document.

Research ethics:

Use of animals and field studies:

It is a condition of publication that you make available the data and research materials supporting the results in the article. Please see our Data Sharing Policies (<https://royalsociety.org/journals/authors/author-guidelines/#data>). Datasets should be deposited in an appropriate publicly available repository and details of the associated accession number, link or DOI to the datasets must be included in the Data Accessibility section of the article (<https://royalsociety.org/journals/ethics-policies/data-sharing-mining/>). Reference(s) to datasets should also be included in the reference list of the article with DOIs (where available).

Please submit a copy of your revised paper within three weeks. If we do not hear from you within this time your manuscript will be rejected. If you are unable to meet this deadline please let us know as soon as possible, as we may be able to grant a short extension.

Best wishes,
 Dr John Hutchinson, Editor
 mailto: proceedingsb@royalsociety.org

Associate Editor
 Board Member: 1
 Comments to Author:

Now, here's a fossil that is not really a looker. A single, highly weathered specimen of something that surely looks like it could be an opabiniid. I was sure that this new specimen surely couldn't offer much new input to the debate about opabiniids, given the much superior material from the Burgess Shale.

However, three reviewers all found the study of notable value. They variably agree with me that this fossil is a bit of an eye sore and some interpretations are hard to interpret. However, as we all know who works with BS type macrofossils, some characters are impossible to illustrate in a publication and even with a single imaging method.

The many sound comments should provide for some reflection for the authors to improve the paper. I suggest it should be considered for publication in Proceedings B. after those have been taken into consideration and perhaps another round of review depending on the response and decisions made during revision.

Reviewer(s)' Comments to Author:

Referee: 1

Comments to the Author(s)

The iconic opabiniid is among the most hotly debated animals from the Cambrian. Only one species, *Opabinia regalis* from the Burgess Shale, has been known for over one century. Here Pates and colleagues reinterpret an old specimen from the Wheeler Formation of Utah as the second representative of opabiniids. The authors further conduct phylogenetic analysis with treespace visualization, revealing that the new named animal being a member of opabiniids. I find this work interesting in general and I agree that this specimen might be opabiniid. But several interpretations in the manuscript, such as mouth opening, extension of setal blades and the number of caudal blades, might need a more cautious manner though.

1) the mouth

The authors interpreted a patch of exposed matrix right behind the eyes, at best with a darker circle outside, as potential mouth of the new opabiniid (Fig 2b). While in *Opabinia regalis*, the mouth is located further back beneath the first trunk segment, primarily due to the backward folding of the esophagus (Whittington 1985). With this ambiguity in structure and difference in position, I hesitate to believe the mouth is preserved.

2) the setal blades

The authors argue that the state of setal blades in Utah opabiniid represent a combination of some radiodonts and *Opabinia* (lines 232-237), they cover the whole dorsal surface and extend to the anterior margin of flap proximal part. I am not so sure about the relative position of setal blades and the flaps. The specimen and interpretative drawing (Fig 1b,c) show that setal blades can extend to the anterior (flaps 10, 11), middle (flaps 5, 6) or even rear margin (flap 9?) of the corresponding flap proximal part. This variation might indicate that the setal blade of Utah opabiniid is more radiodont-like, it does not attach with the flaps.

3) the lobopodous limbs

The interpretation of lobopodous limbs is already quite radical. These slender reddish patches with irregular margin are not well distinct from the flaps (normally broken with irregular

margin). It really drives me into crazy when I saw the indicative segment/annulation lines in the drawings (Fig 1c, 3b), particularly by the one in fig 3b.

4) the number of caudal blades

The author interpreted that Utah opabiniid have at least seven, probably 8 pairs of caudal blades, a much, much larger number in radiodonts and Opabinia. I am not convinced by this interpretation. There is an alternative possibility. When you trace the trunk (marked by seta), you can see that the body axis (which should divide the pairs of the caudal blades) go to the fourth blade counting from the left side. Given the dorsal-lateral compression of the specimen and the overlapping the caudal blades, the one that is most close to us should be exposed as the largest, which should be edged by the two 'caudal ramus'.

In addition, it is always radical to erect new taxa based on single poorly preserved specimen. The sole specimen of Utah opabiniid is nearly complete. However, with the absence of features of proboscis, the only character that differs Utah specimen from Opabinia regalis is the number of caudal blades. It would not be so well-grounded to do taxonomic exercise here.

Referee: 2

Comments to the Author(s)

This paper describes a new fossil opabiniid, a stem-group euarthropod from the Cambrian period. The new taxon is known from a single specimen, and the quality of preservation (while exceptional) leaves some characters in doubt. However, the rarity of opabiniids in the fossil record (this is only the second known species), combined with their pivotal position in the arthropod tree (close to the emergence of the various key characters), amply justifies publication, in my view. The authors have been very honest about the limitations of the fossil material, and take a very measured approach to considering its significance. Indeed, the value of the paper lies as much in its careful methodological approach as in the novelty of the fossil material, and is further enhanced by a detailed exploration of character evolution in this critical part of the arthropod tree. Therefore, I expect the paper to have broad appeal, far beyond specialists in stem-group euarthropods, and the methods could set a new standard for dealing with problematic fossils, and a new way of visualizing support for phylogenetic hypotheses. I recommend the paper for publication with only very minor changes.

Detailed comments:

Placement of new taxon: in general, the worry with a single specimen with partly ambiguous morphology is that a small number of homology assessments can essentially channel the outcome of any phylogenetic analysis. I'm relieved that the authors have faced this potential pitfall head-on by running sensitivity analyses in which the canonical opabiniid character - the 'proboscis' - is coded as uncertain. However, I wonder how sensitive the opabiniid grouping is to a small number of other characters, e.g. the caudal rami (which I find perhaps least convincing of all the described characters). I'm not suggesting that the authors run new sensitivity analyses, but perhaps they could comment on whether they expect this to be a problem. To put it another way, why test the coding of the proboscis, but not other potentially decisive characters?

Abstract and set-up of the radiodont vs. opabiniid comparison: in the abstract (p. 1, lines 20-22) it seems strange to make the point that Anomalocaris and relatives are very diverse but Opabinia (till now) is a one-off. This idea is picked up again in the discussion (p. 18, lines 355-356): "our revised opabiniids have not nearly caught up to the known diversity or distribution of radiodonts". Why would one compare groups like this? It doesn't seem to be meaningful, especially if radiodonts are paraphyletic. I'd be inclined in the abstract to make the point that every new species from this part of the tree has the potential to usefully constrain arthropod character evolution, especially a new representative of a poorly known group. In the discussion (lines 371-374), I find the suggestions of an ecological explanation for relative diversity of radiodonts and opabiniids rather speculative (and again, a bit meaningless if radiodonts are not a

clade). It seems unfair to say that opabiniids show limited evidence for adaptations for different niches, when there's only two of them!

Figures and described anatomy: I'm fascinated by the 'setal blocks' and the inclusion of a magnified image in Fig. 3 is welcome. Do you have even more magnified images that might be included in the supplementary info? (I appreciate the specimen is very small). The reverse imbrication of flaps is a potentially pivotal character, but I don't feel confident that I can see it in the images, and especially not in the interpretative line drawings. Please consider a detailed photo, or a tweak to the line drawing to make this clear.

Minor comments:

Phrase 'hotly contested/hotly debated' sounds rather journalistic - but elsewhere I appreciate that you've tried to explain exactly how the competing hypotheses arise

Lines 79-83: unclear phrasing in "The identification of...": are you talking in general now about what's needed to distinguish plesiomorphies and apomorphies, or are you referring to *Opabinia* in particular?

Line 155 and elsewhere: I find the description of setal blocks extending across the "entire dorsal surface" a bit confusing: you mean across the whole width of the 'trunk' and also the flaps (but "entire" suggests they're also covering the tail and head - ?)

Line 156: "These setal structures..." - you mean the setal blocks taper laterally to a rounded termination? (first I read it as meaning the 'setae' have rounded tips)

Line 266: you refer here and in sup fig 4 to KUMIP 314087, but elsewhere use the new taxon name - is this a deliberate distinction?

Line 317: "related to at least one opabiniid" - but there's only one (*Opabinia*)?

Supp figures 3 and 4 - the trees are too small/low-res for me to read some of the text (especially the support metrics). This may just be in my review copy but please check the final submitted versions are clear.

Referee: 3

Comments to the Author(s)

This manuscript redescribed a fossil specimen KUMIP 314087 from the Wheeler Formation of Utah, which was originally described as an anomalocaridid radiodont. In this manuscript, the authors carried out a careful comparative morphological study and novel phylogenetic and treespace visualization analyses, which suggest that this animal has a close affinity with the Burgess Shale animal *Opabinia*, contributing to our understanding of the diversity and evolution of opabiniids. The manuscript was very well written, the description and analyses are generally robust, and the conclusion and discussion are sound. Therefore, I would like to support its publication on Proceedings B after some minor revisions.

- The structure of the manuscript

1) There are two key novel points in this manuscript: the discovery of an opabiniid from Utah and the novel analytical method of treespace visualization. However, I felt that these two parts are slightly disjointed and lack a good coherency throughout the manuscript. Maybe the authors can emphasize the phylogenetic uncertainty of KUMIP 314087, and then how to use this new method to solve the problem.

2) It is better to have the Material and Method section after the Introduction, so the readers can understand the results better.

- Morphological details

1) It is better to add some explanation of the specimen preservation in the first paragraph of the description before going into morphological details. As the specimen is poorly preserved, some morphological interpretations are not very convincing.

2) Line 142-143: "two curved red structures surround a circular opening, interpreted as a mouth opening". - This structure is not well defined, and the interpretation is not very convincing. It is very close to the possible eyes, and they are all in a similar shape and size. Is it possibly being an

eye but preserved slightly different due to missing the surface? By comparison with Burgess Shale *Opabinia*, the mouth position should be much more ventral and posterior. This structure seems to be too close to the proboscis, which would be very difficult for feeding.

3) Line 144: “interpreted as a pair of lateral eyes” – By comparison with *Opabinia*, I agree that this structure seems to be eye(s), but it is difficult to tell if paired or not.

4) Line 159: “At least 14, likely 15” – I feel that the flaps 11-15 are not completely preserved and miss some parts toward their ventral side margin. The current “flap 14/15” should correspond to the caudal blade 1/2 on the right side.

5) Line 160: “Boundaries are not clear between what are interpreted as the two anteriormost flaps,” – It is true that the boundaries are not clear, but they seem to be clearer in Fig. 1b than in Fig. 1c. The camera-lucida drawing doesn’t match well with the specimen in some details. I wonder if a low-angle light will help to solve the boundaries of these flaps.

6) I barely can see the structures interpreted as lobopodous limbs, which is the least convincing structure in the description.

Overall, I agree that *Utaurora* shows many characters like *Opabinia*, and I have no problem interpreting the proboscis, setal blades, flaps, caudal blades, and caudal rami. Therefore, there is no need to overinterpret some uncertain structures.

- Phylogenetic analyses

1) Please check how the uncertainty of these characters above might influence the phylogenetic results.

2) *Kylinxia* was recently reported from the Chengjiang biota (Zeng et al. 2020), and it shows similar eye morphology with *Opabinia*, and it is also suggested to be at an important phylogenetic position, so please add some comparative discussion with *Kylinxia*, and this taxon should also be included in the phylogenetic analyses.

- Treespace visualization

This is a relatively new method and “the first attempt to use such a visualization to interrogate the distribution of bipartitions for the position of a focal fossil taxon”. Although the authors cited some previous references, it will still be helpful to summarise the methods and principles of this method and explain its advantage compared to traditional methods. Otherwise, I feel that Table 1 is very clear and convincing, and the visualisation in Fig. 4b/c seems unnecessary.

- Others:

Line 55: “the most” change to “one of the most”

Line 79: “remains hotly debated.” – please add relevant references here

Line 121: “of at least 13, likely 15, segments”, in order to be consistent with sentences below, it is better to add “(xx segments in *Opabinia*)”.

Line 138: “(Fig. 1)” change to “(Fig. 1b)”

Author's Response to Decision Letter for (RSPB-2021-2093.R0)

See Appendix A.

RSPB-2021-2093.R1 (Revision)

Review form: Reviewer 1

Recommendation

Accept with minor revision (please list in comments)

Scientific importance: Is the manuscript an original and important contribution to its field?

Excellent

General interest: Is the paper of sufficient general interest?

Excellent

Quality of the paper: Is the overall quality of the paper suitable?

Good

Is the length of the paper justified?

Yes

Should the paper be seen by a specialist statistical reviewer?

No

Do you have any concerns about statistical analyses in this paper? If so, please specify them explicitly in your report.

No

It is a condition of publication that authors make their supporting data, code and materials available - either as supplementary material or hosted in an external repository. Please rate, if applicable, the supporting data on the following criteria.

Is it accessible?

Yes

Is it clear?

Yes

Is it adequate?

Yes

Do you have any ethical concerns with this paper?

No

Comments to the Author

It might be plausible to clarify the interpretation of (the number of) caudal blades with different possibilities. Deformation might work here, but it is really hard to believe that seven or eight caudal blades on the top of the 'caudal rami' have been folded/deformed, or broken into 'buds' with their total size smaller than one of their counterparts.

In addition, re-trace the body axis to the caudal region only reduce the number of caudal blades as four or so, which is still different from *Opabinia*.

This point is not fatal though, and there is no need to go another round of reviewing. I am happy to see this to be published after addressing this issue (or not).

Review form: Reviewer 2

Recommendation

Accept as is

Scientific importance: Is the manuscript an original and important contribution to its field?

Good

General interest: Is the paper of sufficient general interest?

Good

Quality of the paper: Is the overall quality of the paper suitable?

Good

Is the length of the paper justified?

Yes

Should the paper be seen by a specialist statistical reviewer?

No

Do you have any concerns about statistical analyses in this paper? If so, please specify them explicitly in your report.

No

It is a condition of publication that authors make their supporting data, code and materials available - either as supplementary material or hosted in an external repository. Please rate, if applicable, the supporting data on the following criteria.

Is it accessible?

Yes

Is it clear?

Yes

Is it adequate?

Yes

Do you have any ethical concerns with this paper?

No

Comments to the Author

The authors have answered all my queries very satisfactorily. I think they have also responded robustly to the comments of other reviewers. I am happy for the paper to now be published as-is.

Review form: Reviewer 3

Recommendation

Accept as is

Scientific importance: Is the manuscript an original and important contribution to its field?

Good

General interest: Is the paper of sufficient general interest?

Good

Quality of the paper: Is the overall quality of the paper suitable?

Good

Is the length of the paper justified?

Yes

Should the paper be seen by a specialist statistical reviewer?

No

Do you have any concerns about statistical analyses in this paper? If so, please specify them explicitly in your report.

No

It is a condition of publication that authors make their supporting data, code and materials available - either as supplementary material or hosted in an external repository. Please rate, if applicable, the supporting data on the following criteria.

Is it accessible?

Yes

Is it clear?

Yes

Is it adequate?

Yes

Do you have any ethical concerns with this paper?

No

Comments to the Author

I thank the authors for carrying out a careful revision, and the manuscript now has addressed all my previous concerns, so I am happy to support its publication in Proceedings B.

Decision letter (RSPB-2021-2093.R1)

10-Jan-2022

Dear Dr Pates

I am pleased to inform you that your manuscript RSPB-2021-2093.R1 entitled "New opabiniid diversifies the weirdest wonders of the euarthropod stem group" has been accepted for publication in Proceedings B. Congratulations!!

The referee(s) have recommended publication, but also suggest some minor revisions to your manuscript. Therefore, I invite you to respond to the referee(s)' comments and revise your manuscript. Because the schedule for publication is very tight, it is a condition of publication that you submit the revised version of your manuscript within 7 days. If you do not think you will be able to meet this date please let us know. I suggest that you earnestly try to make the final recommended changes.

To revise your manuscript, log into <https://mc.manuscriptcentral.com/prsb> and enter your Author Centre, where you will find your manuscript title listed under "Manuscripts with

Decisions." Under "Actions," click on "Create a Revision." Your manuscript number has been appended to denote a revision. You will be unable to make your revisions on the originally submitted version of the manuscript. Instead, revise your manuscript and upload a new version through your Author Centre.

[http://datadryad.org/submit?journalID=RSPB&manu=\(Document not available\)](http://datadryad.org/submit?journalID=RSPB&manu=(Document%20not%20available)) which will take you to your unique entry in the Dryad repository. If you have already submitted your data to dryad you can make any necessary revisions to your dataset by following the above link. Please see <https://royalsocietypublishing.org/journals/ethics-policies/data-sharing-mining/> for more details.

6) For more information on our Licence to Publish, Open Access, Cover images and Media summaries, please visit <https://royalsocietypublishing.org/journals/authors/author-guidelines/>.

Sincerely,

Dr John Hutchinson

<mailto:proceedingsb@royalsocietypublishing.org>

Associate Editor:

Board Member: 1

Comments to Author:

You will see that referee number 2 still has got some concerns regarding the caudal region and the setal blades, but is overall happy to see the study published.

I suggest in the final submission you decide whether to take these concerns to heart and deliberate more fully on this or revise the study.

Reviewer(s)' Comments to Author:

Referee: 2

Comments to the Author(s)

The authors have answered all my queries very satisfactorily. I think they have also responded robustly to the comments of other reviewers. I am happy for the paper to now be published as-is.

Referee: 1

Comments to the Author(s)

It might be plausible to clarify the interpretation of (the number of) caudal blades with different possibilities. Deformation might work here, but it is really hard to believe that seven or eight caudal blades on the top of the 'caudal rami' have been folded/deformed, or broken into 'buds' with their total size smaller than one of their counterparts.

In addition, re-trace the body axis to the caudal region only reduce the number of caudal blades as four or so, which is still different from *Opabinia*.

This point is not fatal though, and there is no need to go another round of reviewing. I am happy to see this to be published after addressing this issue (or not).

Referee: 3

Comments to the Author(s)

I thank the authors for carrying out a careful revision, and the manuscript now has addressed all my previous concerns, so I am happy to support its publication in Proceedings B.

Decision letter (RSPB-2021-2093.R2)

11-Jan-2022

Dear Dr Pates

I am pleased to inform you that your manuscript entitled "New opabiniid diversifies the weirdest wonders of the euarthropod stem group" has been accepted for publication in Proceedings B.

Data Accessibility section

Open Access

Paper charges

Sincerely,

Proceedings B

Appendix A

Board Member: 1

Comments to Author:

Now, here's a fossil that is not really a looker. A single, highly weathered specimen of something that surely looks like it could be an opabiniid. I was sure that this new specimen surely couldn't offer much new input to the debate about opabiniids, given the much superior material from the Burgess Shale.

However, three reviewers all found the study of notable value. They variably agree with me that this fossil is a bit of an eye sore and some interpretations are hard to interpret. However, as we all know who works with BS type macrofossils, some characters are impossible to illustrate in a publication and even with a single imaging method.

The many sound comments should provide for some reflection for the authors to improve the paper. I suggest it should be considered for publication in Proceedings B. after those have been taken into consideration and perhaps another round of review depending on the response and decisions made during revision.

We thank the editor for their comments, and for arranging the thorough review of this manuscript. We were heartened that all three reviewers provided positive assessments of our work. The major area of concern related to the tentative identification of some parts of the morphology, in particular the possible mouth and lobopodous limbs. We would like to stress that all parts of the morphology only tentatively identified, including mouth orientation and presence of lobopodous limbs, were scored as '?' in our morphological matrix, and thus did not impact on the phylogenetic results.

We have revised our text in response to the reviewers' comments, taking particular care to emphasise areas in which the morphological interpretations are only tentative. We thank all three reviewers for their careful and constructive feedback, which strengthened the manuscript significantly.

We provide a point-by-point response to the reviewer comments below, and outline the changes we have made in the revised text. We have also provided a track-changes version of the manuscript.

Reviewer(s)' Comments to Author:

Referee: 1

Comments to the Author(s)

The iconic opabinid is among the most hotly debated animals from the Cambrian. Only one species, *Opabinia regalis* from the Burgess Shale, has been known for over one century. Here Pates and colleagues reinterpret an old specimen from the Wheeler Formation of Utah as the second representative of opabiniids. The authors further conduct phylogenetic analysis with treespace visualization, revealing that the new named animal being a member of opabiniids. I find this work interesting in general and I agree that this specimen might be opabinid. But several interpretations in the manuscript, such as mouth opening, extension of setal blades and the number of caudal blades, might need a more cautious manner though.

We thank the reviewer for their comments on our manuscript. It is great to hear that the reviewer found the work interesting, and agrees that the specimen might be an opabiniid. We incorporated some of the reviewer's comments and interpretations of the fossil into our revised text, however we maintain that there are enough morphological features present in our proposed holotype to warrant the erection of a new taxon.

1) the mouth

The authors interpreted a patch of exposed matrix right behind the eyes, at best with a darker circle outside, as potential mouth of the new opabiniid (Fig 2b). While in *Opabinia regalis*, the mouth is located further back beneath the first trunk segment, primarily due to the backward folding of the esophagus (Whittington 1985). With this ambiguity in structure and difference in position, I hesitate to believe the mouth is preserved.

The possible mouth opening was only tentatively identified. As this reviewer points out (and supported by reviewer 3) the evidence in support of this interpretation is not strong. Thus we now make more explicit in the main text the extent to which this feature is only very tentatively identified as a possible mouth opening – we feel it is important to offer a *possible* explanation for something which is prominent in the head region. This is more justifiable than the alternative (to not label this feature or to offer no possible morphological interpretation). We also offer an additional possible morphological interpretation (as suggested by reviewer 3) that this might be a poorly preserved eye.

The new sentences read: 'In the ventral posterior region of the head, two curved red structures form an approximately circular outline. This feature could be interpreted as a mouth opening, or alternatively as a poorly preserved eye.' The labelling in figures where this feature is displayed has also been revised.

2) the setal blades

The authors argue that the state of setal blades in Utah opabiniid represent a combination

of some radiodonts and Opabinia (lines 232-237), they cover the whole dorsal surface and extend to the anterior margin of flap proximal part. I am not so sure about the relative position of setal blades and the flaps. The specimen and interpretative drawing (Fig 1b,c) show that setal blades can extend to the anterior (flaps 10, 11), middle (flaps 5, 6) or even rear margin (flap 9?) of the corresponding flap proximal part. This variation might indicate that the setal blade of Utah opabiniid is more radiodont-like, it does not attach with the flaps.

We agree with the reviewer that in some cases the setal blocks appear to cover the middle or even posterior margin of the flap, rather than solely the anterior. However, in all cases the setal blocks do appear to cover not only the width of the trunk but also the proximal parts of the flaps. Thus we have altered our wording in the description, replacing 'anterior' with 'proximal'. This sentence now reads: 'These setal blocks taper to a rounded subtriangular termination, which overlaps the proximal part of the flaps.'

Even considering this broader interpretation, the arrangement of setae in *Utaurora* still represents a difference to *Aegirocassis* and *Peytoia*, where setal blocks are limited to the trunk (they do not appear to cover the flaps, see for example Van Roy et al. 2015, figure 1).

3) the lobopodous limbs

The interpretation of lobopodous limbs is already quite radical. These slender reddish patches with irregular margin are not well distinct from the flaps (normally broken with irregular margin). It really drives me into crazy when I saw the indicative segment/annulation lines in the drawings (Fig 1c, 3b), particularly by the one in fig 3b.

The lobopodous limbs were only ever very tentatively identified. As also pointed out by reviewer 3, there is not strong morphological support for these features. We now label the possible lobopodous limbs as '?' in all figures, and note the alternative explanations for what is observed in the fossil, providing equal weight to three possible interpretations. This part of the description now reads: 'Thin structures protruding from beneath flaps 12 and 13 ("?" in Figs 1, 3) are difficult to interpret. They may represent poorly preserved ventral lobopodous limbs, broken margins of swimming flaps, or be artefacts from the matrix.'

All mention of lobopodous limbs in the Discussion has been removed.

4) the number of caudal blades

The author interpreted that Utah opabiniid have at least seven, probably 8 pairs of caudal blades, a much, much larger number in radiodonts and Opabinia. I am not convinced by this interpretation. There is an alternative possibility. When you trace the trunk (marked by seta), you can see that the body axis (which should divide the pairs of the caudal blades) go to the fourth blade counting from the left side. Given the dorsal-lateral compression of the

specimen and the overlapping the caudal blades, the one that is most close to us should be exposed as the largest, which should be edged by the two 'caudal ramus'.

We agree with the reviewer that the number of caudal blades in *Utaurora* exceeds that of all radiodonts and also *Opabinia*. However, we do not agree with the reviewer's proposed reinterpretation. As outlined in our Figure 3, blades corresponding to the right side of the animal can be seen. Thus the blades on this side have been compressed lateral to the body, and the blades on the left side of the body are the ones that we see splayed out. The total number of identified blades in the figure means that the number of caudal blades in *Utaurora* are higher than in any radiodont or *Opabinia*.

In addition, it is always radical to erect new taxa based on single poorly preserved specimen. The sole specimen of Utah opabiniid is nearly complete. However, with the absence of features of proboscis, the only character that differs Utah specimen from *Opabinia regalis* is the number of caudal blades. It would not be so well-grounded to do taxonomic exercise here.

***Utaurora* differs from *Opabinia* not only in the number of caudal blades, but also in the distinctive arrangement of setal blocks (even with the reviewer's interpretation of dorsal-only setal blades, this differs from *Opabinia*). Thus, given the limited information available, we feel confident that the number of diagnostic characters in our proposed holotype is enough to warrant the erection of a new genus and species.**

Referee: 2

Comments to the Author(s)

This paper describes a new fossil opabiniid, a stem-group euarthropod from the Cambrian period. The new taxon is known from a single specimen, and the quality of preservation (while exceptional) leaves some characters in doubt. However, the rarity of opabiniids in the fossil record (this is only the second known species), combined with their pivotal position in the arthropod tree (close to the emergence of the various key characters), amply justifies publication, in my view. The authors have been very honest about the limitations of the fossil material, and take a very measured approach to considering its significance. Indeed, the value of the paper lies as much in its careful methodological approach as in the novelty of the fossil material, and is further enhanced by a detailed exploration of character evolution in this critical part of the arthropod tree. Therefore, I expect the paper to have broad appeal, far beyond specialists in stem-group euarthropods, and the methods could set a new standard for dealing with problematic fossils, and a new way of visualizing support for phylogenetic hypotheses. I recommend the paper for publication with only very minor changes.

We thank the reviewer for their positive comments on our manuscript. In particular, we appreciate that the reviewer has noted our honesty relating to the limitations of the fossil material, and the value of incorporating a treespace analysis in the study. We too hope and expect the paper to have a broad appeal.

Detailed comments:

Placement of new taxon: in general, the worry with a single specimen with partly ambiguous morphology is that a small number of homology assessments can essentially channel the outcome of any phylogenetic analysis. I'm relieved that the authors have faced this potential pitfall head-on by running sensitivity analyses in which the canonical opabiniid character – the 'proboscis' – is coded as uncertain. However, I wonder how sensitive the opabiniid grouping is to a small number of other characters, e.g. the caudal rami (which I find perhaps least convincing of all the described characters). I'm not suggesting that the authors run new sensitivity analyses, but perhaps they could comment on whether they expect this to be a problem. To put it another way, why test the coding of the proboscis, but not other potentially decisive characters?

We thank the reviewer for highlighting the value of the sensitivity analysis. Other poorly preserved features (e.g. mouth orientation, possible lobopods, possible eyes) were coded as '?', and so these will not impact on the phylogenetic analysis. These are discussed further in the supplemental text.

We have added a brief explanation of this coding strategy to the Materials and Methods: 'Anatomical features that were only tentatively identified for KUMIP 314087 were coded as '?'. In the case of the proboscis, owing to its uniqueness in *Opabinia* and its relevance to the discussion of the affinity of *Utaurora comosa*, two matrices were

generated, one coding this character as present and the other as “?” (further details in Supplementary Material). ‘

We chose to run the sensitivity analysis on the proboscis character for two reasons. Firstly, it is a feature which is very well known in *Opabinia* and no other terminal in the matrix, so it might be expected to shape the analysis. Secondly, it is not very well preserved in *Utaurora*, so it was important to test that our results (*Utaurora* is an opabiniid) did not solely rely on this interpretation. We consider that the caudal rami are much better preserved than the proboscis, and so did not deem it necessary to run further sensitivity analyses on these features. However, we appreciate that the reviewer does not find these rami as convincing as we do.

To explore the reviewer’s suggestion, we recoded character 109 (spines on caudal rami) as uncertain for *Utaurora* and ran only the Bayesian inference analysis using the maximum information strategy (due to speed of analysis. The result highlights monophyletic opabiniids below (in pink). The sensitivity analysis has performed similarly to the recoded proboscis where posterior probability favoring opabiniid monophyly decreased to 0.58; here, posterior probability support is decreased to 0.54 but still slightly favoring monophyly of opabiniids.

Due to the long analysis time required for the minimum assumptions strategy, we have not fully reanalyzed this sensitivity analysis but have modified the text to reflect the impact of individual uncertain character states in *Utaurora*. The last sentence of this

subsection now reads: ‘The sister group relationship of *Utaurora* with *Opabinia* (rather than radiodonts or deuterochordates) is not driven solely by the proboscis character, and is maintained due to a suite of shared morphological characters (e.g. dorsal furrows, caudal rami and proboscis).’

Abstract and set-up of the radiodont vs. opabiniid comparison: in the abstract (p. 1, lines 20-22) it seems strange to make the point that *Anomalocaris* and relatives are very diverse but *Opabinia* (till now) is a one-off. This idea is picked up again in the discussion (p. 18, lines 355-356): “our revised opabiniids have not nearly caught up to the known diversity or distribution of radiodonts”. Why would one compare groups like this? It doesn’t seem to be meaningful, especially if radiodonts are paraphyletic. I’d be inclined in the abstract to make the point that every new species from this part of the tree has the potential to usefully constrain arthropod character evolution, especially a new representative of a poorly known group. In the discussion (lines 371-374), I find the suggestions of an ecological explanation for relative diversity of radiodonts and opabiniids rather speculative (and again, a bit meaningless if radiodonts are not a clade). It seems unfair to say that opabiniids show limited evidence for adaptations for different niches, when there’s only two of them!

The comparison between radiodonts and opabiniids is partly historical – both were described around the same time, placed into the stem group of euarthropods (sometimes together in the clade Dinocarida e.g. Collins 1996), and display morphological combinations that Gould identified as ‘weird wonders’. We appreciate the reviewer’s point, that emphasising the importance of these taxa in constraining euarthropod evolution is important.

We have altered the abstract to highlight the importance of *Anomalocaris* and *Opabinia* in constraining the polarity of key euarthropod characters, and we have removed the last paragraph in the discussion subsection that dealt with the ecological interpretations, so that the main focus is on the historical comparison.

We would also emphasise that this manuscript does not provide strong support for radiodont paraphyly, as discussed in the supplementary information.

Figures and described anatomy: I’m fascinated by the ‘setal blocks’ and the inclusion of a magnified image in Fig. 3 is welcome. Do you have even more magnified images that might be included in the supplementary info? (I appreciate the specimen is very small). The reverse imbrication of flaps is a potentially pivotal character, but I don’t feel confident that I can see it in the images, and especially not in the interpretative line drawings. Please consider a detailed photo, or a tweak to the line drawing to make this clear.

Unfortunately we have provided the highest magnification images of these setal blocks that are available to us in the main document. The reverse imbrication is most visible around midway down the trunk (flaps 6-10). We have tweaked the line drawing in Figure 1 to make this clearer (flap boundaries that sit underneath other flaps are now dotted), and mentioned it in the text.

Minor comments:

Phrase 'hotly contested/hotly debated' sounds rather journalistic – but elsewhere I appreciate that you've tried to explain exactly how the competing hypotheses arise

We have removed these two phrases. We have replaced the first with 'controversial'. The sentence containing 'hotly debated' has been removed in the revised text.

Lines 79-83: unclear phrasing in "The identification of...": are you talking in general now about what's needed to distinguish plesiomorphies and apomorphies, or are you referring to *Opabinia* in particular?

We intended the former – what's needed to distinguish plesiomorphies and apomorphies generally within stem groups. We have therefore altered the sentence, which now reads: 'The identification of plesiomorphic and apomorphic characters within the euarthropod stem lineage has required...'

Line 155 and elsewhere: I find the description of setal blocks extending across the "entire dorsal surface" a bit confusing: you mean across the whole width of the 'trunk' and also the flaps (but "entire" suggests they're also covering the tail and head - ?)

We thank the reviewer for this suggested alteration in our wording. We have changed 'entire dorsal surface' to 'whole width of the trunk' in all three places.

Line 156: "These setal structures..." - you mean the setal blocks taper laterally to a rounded termination? (first I read it as meaning the 'setae' have rounded tips)

We agree that the wording could be clearer here. We have tweaked the sentence, which now reads: 'These setal blocks taper to a rounded subtriangular termination...'

Line 266: you refer here and in sup fig 4 to KUMIP 314087, but elsewhere use the new taxon name – is this a deliberate distinction?

We thank the reviewer for pointing this out, this was not a deliberate distinction. We have changed it to the new taxon name in both places.

Line 317: "related to at least one opabiniid" – but there's only one (*Opabinia*)?

We thank the reviewer for noticing this, we have altered the sentence to 'that support *Utaurora* forming a clade with *Opabinia*, and not with an alternative taxon'.

Supp figures 3 and 4 – the trees are too small/low-res for me to read some of the text (especially the support metrics). This may just be in my review copy but please check the final submitted versions are clear.

Thank you for highlighting this, we will check in the final versions that the resolution is high enough to read the text and support metrics. We will also submit high resolution versions of all figures as supplemental documents.

Referee: 3

Comments to the Author(s)

This manuscript redescribed a fossil specimen KUMIP 314087 from the Wheeler Formation of Utah, which was originally described as an anomalocaridid radiodont. In this manuscript, the authors carried out a careful comparative morphological study and novel phylogenetic and treespace visualization analyses, which suggest that this animal has a close affinity with the Burgess Shale animal Opabinia, contributing to our understanding of the diversity and evolution of opabiniids. The manuscript was very well written, the description and analyses are generally robust, and the conclusion and discussion are sound. Therefore, I would like to support its publication on Proceedings B after some minor revisions.

We thank the reviewer for their positive comments on our manuscript. It was heartening to read that the reviewer considers the study generally robust, with a sound discussion and conclusion. We appreciate the reviewer's comments, and have made changes to the text relating to uncertainty in identifying some morphological features in the material. We would like to highlight that these uncertain morphological features did not impact on the phylogenetic analyses, as we coded them as uncertain in the matrix.

- The structure of the manuscript

1) There are two key novel points in this manuscript: the discovery of an opabiniid from Utah and the novel analytical method of treespace visualization. However, I felt that these two parts are slightly disjointed and lack a good coherency throughout the manuscript. Maybe the authors can emphasize the phylogenetic uncertainty of KUMIP 314087, and then how to use this new method to solve the problem.

This structure is what we were aiming for with our first Discussion paragraph: *The power of treespace for phylogenetic uncertainty of fossils*. We have edited the first paragraph of this subsection to expand on this topic, and make this link between treespace and KUMIP 314087 clearer.

2) It is better to have the Material and Method section after the Introduction, so the readers can understand the results better.

We have moved the Materials and Methods to be after the Introduction.

- Morphological details

1) It is better to add some explanation of the specimen preservation in the first paragraph of the description before going into morphological details. As the specimen is poorly preserved, some morphological interpretations are not very convincing.

We thank the reviewer for this input. We have made changes relating to morphological interpretations deemed not convincing (below). We have added a short explanation of the preservation in the first paragraph: 'The head and anterior of the trunk are imperfectly

preserved; however, fine morphological details can be observed in most of the trunk and tail fan.'

2) Line 142-143: "two curved red structures surround a circular opening, interpreted as a mouth opening".—This structure is not well defined, and the interpretation is not very convincing. It is very close to the possible eyes, and they are all in a similar shape and size. Is it possibly being an eye but preserved slightly different due to missing the surface? By comparison with Burgess Shale Opabinia, the mouth position should be much more ventral and posterior. This structure seems to be too close to the proboscis, which would be very difficult for feeding.

We only ever tentatively identified this structure as a possible mouth, however we agree that it could represent a poorly preserved eye. We have altered this part of the description to reflect this possible alternative, and to highlight the tentative nature of our identification (also discussed in our response to Reviewer 1). It now reads: '...two curved red structures form an approximately circular outline. This feature could be interpreted as a mouth opening, or alternatively as a poorly preserved eye.' The figures have also been updated.

3) Line 144: "interpreted as a pair of lateral eyes"—By comparison with Opabinia, I agree that this structure seems to be eye(s), but it is difficult to tell if paired or not.

We agree that it is difficult to tell if this structure is paired or not. We have altered the description to reflect this uncertainty. It now reads: 'This possible eye or mouth opening is immediately proximal to a dark red region of one or two oval shapes, tentatively interpreted as one or two lateral eyes.'

4) Line 159: "At least 14, likely 15"—I feel that the flaps 11-15 are not completely preserved and miss some parts toward their ventral side margin. The current "flap 14/15" should correspond to the caudal blade 1/2 on the right side.

We thank the reviewer for highlighting this. We have added the sentences: 'The posterior flaps (fl13-15) are not completely preserved, especially the ventral margin. A small triangle of setal block present on the anterior margin of the posteriormost flap ("fl15" in Fig. 1) distinguishes this flap from the caudal blades. However, if instead this setal block is considered to associate with the posterior margin of flap 14, then flap 15 should be treated as part of the caudal fan.'

5) Line 160: "Boundaries are not clear between what are interpreted as the two anteriormost flaps,"—It is true that the boundaries are not clear, but they seem to be clearer in Fig. 1b than in Fig. 1c. The camera-lucida drawing doesn't match well with the specimen in some details. I wonder if a low-angle light will help to solve the boundaries of these flaps.

The overlap of features and poor preservation of flap boundaries at the anterior introduced uncertainty at the front of the specimen. This is why we indicated in the text that although we interpret two flaps here, the boundaries are not clear, especially when compared to flaps that sit further back in the specimen. We have tweaked the line drawing (fig 1c) for the anteriormost flap to make it slightly clearer. The possible anterior margin is now included as a dotted line.

6) I barely can see the structures interpreted as lobopodous limbs, which is the least convincing structure in the description.

Following this suggestion, and a similar one by reviewer 1, we have now significantly altered this part of the description (it now reads: Thin structures protruding from beneath flaps 12 and 13 (“?” in Figs 1, 3) are difficult to interpret. They may represent poorly preserved ventral lobopodous limbs, broken margins of swimming flaps, or be artefacts from the matrix). All other mention of lobopodous limbs in the manuscript have been removed.

Overall, I agree that *Utaurora* shows many characters like *Opabinia*, and I have no problem interpreting the proboscis, setal blades, flaps, caudal blades, and caudal rami. Therefore, there is no need to overinterpret some uncertain structures.

We thank the reviewer for highlighting areas in which they believe we over interpreted the specimen. It is our intention to get as much meaningful information from the fossil, while being clear about the limitations of the material. Our revised text more clearly distinguishes between well preserved morphological features, and those which are only tentative (and not included in the phylogenetic analyses).

- Phylogenetic analyses

1) Please check how the uncertainty of these characters above might influence the phylogenetic results.

Fortunately, we have already conservatively coded the features mentioned above (mouth opening, character 48; presence of lobopodous limbs, character 16; presence of eyes, character 55) as ‘?’ in the matrix, therefore these cannot contribute to grouping *Utaurora* with *Opabinia*. The number of caudal blades (character 105), and setal blade organization (character 83) are both autapomorphies of *Utaurora* so they also cannot contribute to this grouping. The exact number of lateral flaps is not a character in our matrix. Thus we can be sure that uncertainty relating to these morphological features is not impacting on our phylogenetic results. For more information, see section *Scoring strategy for tentatively identified features of Utaurora comosa* in the supplementary materials. We also now briefly explain this coding strategy in the Materials and Methods section: ‘Anatomical features that were only tentatively identified for KUMIP 314087 were coded as ‘?’. In the case of the proboscis, owing to its uniqueness in *Opabinia* and its relevance to the

discussion of the affinity of *Utaurora comosa*, two matrices were generated, one coding this character as present and the other as “?” (further details in Supplementary Material).

2) *Kylinxia* was recently reported from the Chengjiang biota (Zeng et al. 2020), and it shows similar eye morphology with *Opabinia*, and it is also suggested to be at an important phylogenetic position, so please add some comparative discussion with *Kylinxia*, and this taxon should also be included in the phylogenetic analyses.

We agree that *Kylinxia* is a very interesting taxon, and that it has been suggested to occupy an important phylogenetic position. However, *Kylinxia* has been proposed as a possible sister group to all other deuterozoans. Thus, we did not expect the inclusion of additional deuterozoans such as *Kylinxia* to impact significantly on our topology. To test this, we have run an additional analysis which includes *Kylinxia* within our matrix. We followed Zeng et al. (2020, their supplementary info “Affinities of frontalmost appendages in Radiodonta and Megacheira”) in interpreting the frontal appendage of *Kylinxia* as deutocerebral.

We ran this additional analysis using the ‘maximum information’ Bayesian inference strategy. Comparison with our results shows that the inclusion of *Kylinxia* does not impact on our results (see opabiniids highlighted in pink and *Kylinxia* in blue, below). The position of *Kylinxia* is likely driven by the interpretation of deutocerebral frontal appendages (in our dataset shared with megacheirans), but again, this initial interpretation follows Zeng et al. (2020).

We chose not to rerun all of our analyses again, as these take several weeks to run (in particular the ‘minimum assumptions’ analysis and subsequent multidimensional treespace). This would have taken us beyond the deadline for resubmission. Thus, as the

results of this additional analysis and the one in the original manuscript are so similar, we do not feel that this is necessary.

While the number of eyes in *Kylinxia* is comparable to *Opabinia*, uncertainty about the presence and number of eyes in the head region of *Utaurora* (and our coding of the eyes as uncertain, as mentioned above and in supplementary text) precludes a meaningful comparison between these two taxa. The remainder of the body of *Kylinxia* is arthrodized and bears arthropodized biramous limbs – it is very different from *Utaurora*. Thus while a comparative discussion between *Kylinxia* and *Opabinia* is warranted, based on shared characters in the head region (and provided in Zeng et al. 2020), one between *Kylinxia* and *Utaurora* is harder to justify.

- Treespace visualization

This is a relatively new method and “the first attempt to use such a visualization to interrogate the distribution of bipartitions for the position of a focal fossil taxon”. Although the authors cited some previous references, it will still be helpful to summarise the methods and principles of this method and explain its advantage compared to traditional methods. Otherwise, I feel that Table 1 is very clear and convincing, and the visualisation in Fig. 4b/c seems unnecessary.

We have added additional details to both the Introduction and Discussion, while the move of the Materials and Methods to be after the Introduction has also made this methodology more explicit in the manuscript.

Table 1 and Fig. 4b/c are complementary. Table 1 is informative with regards to quantitative support for specific bipartitions of interest, while Fig. 4b/c visualises this in multidimensional space. The visualisation provides additional information over the raw numbers in Table 1. For example, it demonstrates the presence of a number of tree islands supporting different bipartitions with relatively little overlap between opabiniids forming a clade with deuteropods (both purple colors in Fig. 4b) vs not forming a clade with deuteropods. Visually, the area of the multidimensional space occupied by the pink dots (supporting *Opabinia* + *Utaurora*) is clearly larger than any of the alternative bipartitions. Fig. 4c allows us to demonstrate that the differences in variation between the two Bayesian parameter sets used have a relatively small impact on the topologies sampled, because they occupy a similar amount of axis 1 and about 2/3 the same amount of axis 2. This is actually less obvious from Table 1 alone. Thus the inclusion of both Table 1 and Fig 4b/c is not redundant.

- Others:

Line 55: “the most” change to “one of the most”

Arthropods are the most diverse phylum in terms of number of species – as supported by reference [1] in the main text, so we kept this sentence as written.

Line 79: “remains hotly debated.” —please add relevant references here

We have chosen to remove this sentence as it does not add significantly to the discussion.

Line 121: “of at least 13, likely 15, segments”, in order to be consistent with sentences below, it is better to add “(xx segments in *Opabinia*)”.

We have added ‘(15 in *Opabinia*)’, as suggested.

Line 138: “(Fig. 1)” change to “(Fig. 1b)”

Changed, as suggested.